# Classification, Distribution, Biosynthesis, and Regulation of Secondary Metabolites in *Matricaria chamomilla*

**Hanbin Wu, Ke Yang, Liwei Dong, Jiabao Ye \* and Feng Xu \***

College of Horticulture and Gardening, Yangtze University, Jingzhou 434025, China
\* Correspondence: yejiabao@yangtzeu.edu.cn (J.Y.); xufeng@yangtzeu.edu.cn (F.X.)

**Abstract:** *Matricaria chamomilla* is a multi-use aromatic medicinal plant, and is known to be one of the oldest medicinal plants in the world. *M. chamomilla* contains abundant volatile oils, of which terpenes and flavonoids are the main medicinal active ingredients, including chamazulene and α-bisabolol. *M. chamomilla* is often used to treat flatulence, inflammation, and other disorders. It is also used for pain relief and sedation. In recent years, many studies have examined the medicinally active ingredients, pharmacological efficacy, plant physiology, and other aspects of *M. chamomilla*. Here, we summarize studies on the secondary metabolites of medicinally active components in *M. chamomilla*, with respect to their biosynthesis pathways and regulation.

**Keywords:** *Matricaria chamomilla*; volatile oil; medicinal active ingredients; terpenoids; flavonoids

## 1. Introduction

*Matricaria chamomilla* is an ornamental medicinal aromatic plant of the composite family Asteraceae, widely distributed in Europe, Asia, the Mediterranean, Southern Africa, and Northwestern America. As a result of its unique fragrance and pharmacological effects, it is widely used in food, beverages, tobacco, daily chemicals, medicine, and other fields, with good market development potential for new agricultural plants [1]. There are two species of chamomile that are commonly used: *Chamaemelum nobile* and *M. chamomilla*, derived from the family Asteraceae [2]. Because the appearance of the two is very similar and the standard of medicinal materials is low, *C. nobile* is often mistaken for *M. chamomilla* in the process of purchasing medicinal materials. According to a large number of pharmacological studies on the volatile oil of *M. chamomilla*, the plant is a medicinal herb with antiseptic, anti-inflammatory, antispasmodic, and soothing effects [3]. In recent years, researchers at home and abroad have focused on the validation of its pharmacological and clinical effects and the development of related products. This paper reviews the types of terpenoids and flavonoid in *M. chamomilla*; the synthesis pathways, regulatory factors, and functions of secondary metabolites; and the identification, regulatory mechanisms, and effects of biosynthesis-related genes. We also explore the effects of exogenous hormones, biotic and abiotic stress, geographical environment, and other factors on the secondary metabolites and content of *M. chamomilla*. We hope to lay a foundation for further research and utilization of *M. chamomilla*.

## 2. Biological Characteristics

*M. chamomilla* is a herbaceous plant that is wild in field heaths, roadsides, and river valleys, and it is native to temperate regions of Europe and Western Asia. It is highly adaptable to soil and the environment, is relatively hardy, and can even grow on alkaline lands at altitudes of 2000 m and at pH 8.0–9.0 [4]. However, the most suitable growing environment for *M. chamomilla* is dry soil at 30 °C–32 °C and air with 40–50% relative humidity [5,6]. Depending on the regional environmental factors, the sowing and harvesting seasons also differ. Winter sowing is an option in tropical and subtropical plains, early in October each

year. Flowering occurs from February to April of the next year. Summer sowing is suitable for high-altitude areas, and the flowering period is from May to June [4]. *M. chamomilla* is an annual herb, about 30–40 cm in length, with a straight, smooth, hairless stem and many branches. The leaf blades (2–3 alternate) are pinnately divided, short, and sessile. The herb has heteromorphic inflorescence, and at the top of the stems, it has a corymbose arrangement that grows on the branches and leaf axils, with a pedicel. The involucrum bract of the flower is hemispherical, and the bract margin has two membranes. A layer of white tongue-like peanuts has also been observed on the periphery of inflorescence. The apex is flat or slightly concave, and the corolla hangs low after blooming. The inner layer is an amphoteric yellow tubular flower, and the four to five branches of the corolla have toothed lobes. The stamens consist of five polydrugs. The anther base is rounded and blunt, and the pistil stigma has two splits. The smell is sweet, fruity, and slightly bitter. The fruit is off-white and 1–1.5 mm in length. The weight of 1000 seeds is 0.026–0.03 g [7,8].

## 3. Active Constituents of *M. chamomilla*

The previous studies have found that *M. chamomilla* contains more than 120 types of medicinal active ingredients, including 56 types of organic acids, 36 types of flavonoids, 28 types of terpenoids, and other compounds such as coumarins [5]. Yang and Pan [9] identified two flavonoids, lutein-7-O-β-D-glucoside (II) and apigenin-7-O-β-D-glucoside (I), in the inflorescence of *M. chamomilla* from Xinjiang. Zhou and Li [10] found that *M. chamomilla* contains quercetin, apigenin, luteolin, umbelliferone, and luteolin 7-O-β-d-glucoside (V). In addition, galactose, galacturonic acid, xylose, and choline were detected in *M. chamomilla*. Apigenin is an abundant water-soluble flavonoid in *M. chamomilla*. It mainly exists in the form of apigenin-7-O glucoside and other acylated derivatives [11].

As the main medicinal active ingredient source of volatile oil, the most important substances are terpenoids. Monoterpenes and sesquiterpenes are the most abundant, of which oxygen-containing derivatives are less abundant but have an aroma. In recent years, researchers used GC–MS to analyze the volatile oil components of *M. chamomilla*. They found that the main components in the volatile oil of *M. chamomilla* are sesquiterpenoids, such as α-bisabolol, chamazulene, α-bisabolol oxide, farnesene, β-ocimene, α-elemene, α-pinene, and absinthol [12]. Sesquiterpene, flavonoids, coumarins, and polyacetylene are considered the most important substances in the medicinal active ingredients of *M. chamomilla* [13].

The volatile oil of the head-shaped inflorescence, leaves, and whole plants can be extracted via water vapor distillation and assayed using GC/MS, but the content of bisabolol oxide varies between different varieties [14,15]. During the extraction process, the contents of components in fresh and dried inflorescences were different, and the content of volatile oil was the highest when the tongue-shaped corolla was expanded horizontally [16]. Solid-phase microextraction and headspace solid-phase microextraction combined with fast GC can be used as a supplement or substitute for SD-F-GC in the analysis of volatile components in the Steam distillation of *M. chamomilla* [17]. GC-TOF/MS determination of the main components of chamomile from different areas had little difference, but the relative content of the same component in their essential oils was different [18].

## 4. Terpenoids

### 4.1. Types and Functions of Terpenoids in M. chamomilla

The volatile oil of *M. chamomilla* contains monoterpene cineole, limonene, α-pinene, terpineol, sesquiterpene chamazulene, caryophyllene, farnesene, bisabolol, and oxides. α-bisabolol, α-bisabolol oxides, (E)-β-farnesene, chamazulene, and (Z)-ene-dicycloether are the main medicinal ingredients [19]. Sesquiterpene derivatives are particularly abundant in the inflorescence, among which the effective ones are chamazulene α-bisabolol and oxides (α-bisabolol oxide A and B) [11,20,21]. α-Bisabolol is an important commercial additive; it is a colorless or yellowish low-toxicity sesquiterpene with a unique sweetness and aroma. It is insoluble in water but soluble in ethanol, fats, and oils [22]. It can induce

apoptosis of cancer cells, has an antiproliferative effect on pancreatic cancer cell lines [23], and inhibits glioblastoma cell migration and invasion by downregulating c-Met [24]. $\alpha$-Bisabolol also has antibacterial, anti-inflammatory, antispasmodic, sedative, analgesic, antiseptic, antioxidant, skin soothing, and other effects; it is also an important additive for cosmetics and pharmaceutical products [25].

*M. chamomilla* essential oil is dark blue because it contains chamazulene, a sesquiterpene of guaiacum, which can turn green or even brown over time. Chamazulene is a sesquiterpene lactone, which is the product of the reaction of pre-chamazulene during distillation. It has significant anti-inflammatory, antispasmodic, antibacterial, and bacteriostatic activities, and can also scavenge free radicals [11,26,27]. Chamazulene is not only one of the most valuable components, but it is also the standard for assessing the quality of *M. chamomilla* herbs. β-Caryophyllene and β-farnesene are known as stress regulators that protect plants from herbivorous attacks and environmental stress, whereas coumarins have antispasmodic and antimicrobial properties [21,28,29].

*4.2. Research Progress in Terpene Biosynthesis*

The volatile secondary metabolites of plant terpenoids have important biological and ecological functions, which are mainly expressed in the interaction between plant and environment and the transfer of biological information [30,31] (Figure 1). The synthesis of volatile terpenoids of *M. chamomilla* is tissue-specific, and they are mainly released in flower organs and leaves. Zhang et al. [32] used the Kyoto Encyclopedia of Genes and Genomes to analyze the roots, stems, leaves, and flower tissues of *M. chamomilla* to construct a transcriptome library. They found that the genes involved in the sesquiterpene synthesis pathway are expressed differently in varying tissues, and these differential expression patterns of terpene genes may lead to differences in the type and content of sesquiterpene in various organs. The expression of genes involved in the mevalonic acid (MVA) pathway is higher in leaves and stems than in roots and flowers. Acetyl-CoA C acetyltransferase (AACT), 3-hydroxy-3 methyl glutaryl coenzume A reductase (*HMGR*1, *HMGR*2, and *HMGR*3), mevalonate kinase (MK), phosphomevalonate kinase (*PMK*), and isopentenyl diphosphate-isomer isomerase (IPPI) were significantly more highly expressed in flowers than in other organs, indicating a positive correlation with the content of total sesquiterpenes. Thus, the flower is the key organ for the synthesis of sesquiterpenes in *M. chamomilla* [32]. Previous studies on *Litsea cubeba* have also shown that *AACT*, *HMGR*, 3-hydroxy-3-methylglutaryl-CoA synthase (*HMGS*), phosphate mevalonate kinase (PMVK), the MVA kinase gene (MVK), and mevalonate diphosphate decarboxylase (MVD) are more expressed in the flowers than in other parts [33]. Farnesyl pyrophosphate synthase (FPPS) catalyzes dimethylallyl diphosphate, with two isoprene diphosphate molecules condensing to form farnesyl pyrophosphate (FPP). FPP is a precursor to all sesquiterpenes, and the leaves act as the main synthetic organ for $\alpha$-farnesene and germacrene D [34]. FPPS and geranylgeranyl diphosphate synthase (GGPPS) exhibit the same expression pattern in *M. chamomilla*, with the highest levels of expression in the leaves, followed by those in the flowers, roots, and stems. These data suggested that the organ-specific expression of FPPS and GGPPS regulates the biosynthesis of $\alpha$-farnesene and germacrene D [32]. Cytochrome P450 (CYP450) is a key enzyme in the oxidation modification of plant terpenes; it catalyzes the specific oxidation of the spatial structure of triterpenoid skeletons to form functional groups, such as hydroxyl, carbonyl, and carboxyl groups [35]. Cytochrome P450 reductase (CPR) is the key enzyme gene for the synthesis of chamazulene; it is expressed differently in various organs of *M. chamomilla*, especially in the corolla, and increasing the expression level of this gene can effectively increase the content of chamazulene in *M. chamomilla* [36].

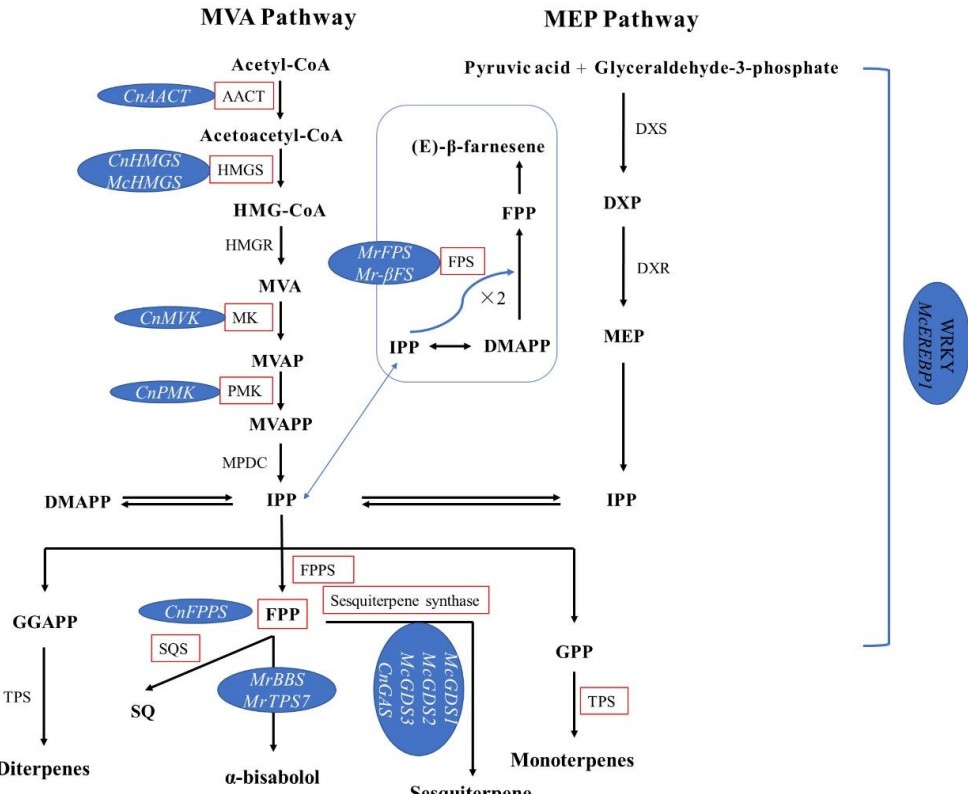

**Figure 1.** Terpenoid synthesis pathway in *M. chamomilla*. Enzyme abbreviations: AACT (acetyl-CoA C-acetyltransferase); DMAPP (dimethylallyl diphosphate); DXP (1-deoxy-xylose-5-phosphate); DXS (1-deoxy-D-xylulose 5-phosphate synthase); FPP (farnesyl pyrophosphate); FPPS (farnesyl pyrophosphate synthase); GGAPP (geranylgeranyl pyrophosphate); GPP (geranyl diphosphate); HMGR (3-hydroxy-3-methylglutaryl-CoA reductase); HMGS (3-hydroxy-3-methylglutaryl-CoA synthase); IPP (isoprene diphosphate); MEP (2-C-methyl-D-erythritol 4-phosphate); MK (MVA kinase); MPDC (diphospho-MVA decarboxylase); MVA (mevalonate); MVAP (mevalonate-5-pyrophosphate); MVAPP (mevalonate 5-diphosphate); PMK (phosphomevalonate kinase); SQ (squalene); SQS (squalene synthase); TPS (terpene synthase).

*AACT* is highly conserved in molecular evolution and is a key gene for regulating the biosynthesis of terpenoids. Fu et al. [37] cloned the *CnAACT* gene, which is highly homologous to the *AACT* gene in other plants. The gene is essential for *M. chamomilla* to produce terpenoids and isoprenoids, such as sterols, carotenoids, and growth regulators. *PMK* and *MVK* are key rate-limiting enzymes in sesquiterpene biosynthesis that are evolutionarily conserved. *CnPMK* and *CnMVK* are homologous to *PMK* and *MVK* from other plants [38,39], and their expression may directly or indirectly affect the content of terpenoids in *M. chamomilla*. The exogenous hormones methyl jasmonate (MeJA) and salicylic acid (SA) also induce sesquiterpene biosynthesis and upregulate the expression of the *CnHMGS* gene, suggesting that *CnHMGS* may be involved in the signaling molecules' responses to environmental stimuli [40]. The expression levels of the *McHMGS* gene in the flower stem and MeJA treatment were consistent with the results of Cheng et al. [40], and the *McHMGS* gene restored the *HMGS*-defective yeast mutant to normal [41,42]. MeJA can also induce the expression of FPPS and (E)-b alkyne synthase, improving the biosynthesis of sesquiterpenes [41–43]. Numerous studies showed that the content of sesquiterpenoids can be increased by increasing the level of the *HMGR* gene in plants [44], and the specific expression of the *CnHMGR* gene in flower stems indicates that it may be involved in the upregulation of sesquiterpene biosynthesis [44].

Sun [45] obtained 9 FPPS genes associated with 15 volatile terpenoids (5 monoterpenes and 10 sesquiterpenes) through co-expression analysis. They are relatively conservative in

the process of plant molecular evolution and have high homology with the sequence of *Artemisia carvifolia*, which is consistent with the results of Tai [46] and Yang [47]. Yang [47] established a rapid propagation system of *M. chamomilla* through plant tissue culture and successfully constructed the plant expression vector *pBI*121-FPS, which was transformed into *Nicotiana tabacum* to obtain resistant tobacco plants. Their work provided a theoretical basis and technical support for controlling the content of sesquiterpenoids such as *M. chamomilla* chamazulene and establishing a genetic transformation system of *M. chamomilla*. FPS was expressed differently in different organs, and its recombinant FPS protein was localized in chloroplasts [45,47]. The expression of *MrFPS* was different in tissues, especially in the ligulate flower and leaf, but not in the stem and tubular flower. FPS expression levels were detected in *Withania kasuensis* [48], *Hedychium coronarium* [49], and *Chimonanthus praecox* [50] in association with the specific accumulation of sesquiterpene. We speculate that strong expression of FPS genes may lead to the specific accumulation of volatile sesquiterpene during flower blooming, indirectly regulating the synthesis of sesquiterpene in *M. chamomilla* flowers. After treatment with 100 $\mu mol \cdot L^{-1}$ MeJA, the expression of the FPS gene increased significantly and reached its highest level after 12 h [43]. βFS (β-Farnesene synthase) has (E)-β-farnesene activity, which is homologous to *Artemisia annua* βFS and can catalyze FPP to generate a single sesquiterpene (E)-β-farnesene. After 100 $\mu mol \cdot L^{-1}$ MeJA treatment for 24 h, the biosynthesis of sesquiterpene (E)-β-farnesene increased, which was positively correlated with the accumulation of sesquiterpene. After MeJA induction, significant differences were found in the contents of five volatile terpenoids (germacrene D, β-ocimene, γ-elemene, (E)-β-farnesene, and α-farnesene) in the leaves of *M. chamomilla* between the controls [51].

α-Bisabolol and its oxides are present only in buds and flowers, but there is an inverse correlation in concentration between the two. When the content of α-bisabolol increased significantly, the concentration of α-bisabolol oxide A was relatively low. α-Bisabolol is a typical sesquiterpene of *M. chamomilla* with anti-inflammatory activity, and α-bisabolol synthase is one of the keys to the biosynthesis of α-bisabolol [52]. To study the transcriptional abundance of terpene synthase and its products in different organs of *M. chamomilla*, Irmisch et al. [53] cloned five terpene synthase (TPS) genes, including four sesquiterpene synthase and one monoterpene synthase. The main products were (E)-β-caryophyllene, germacrene D, isobutene (α and β), β-elemene (E or Z), and β-ocimene, respectively. In contrast to the aboveground organs of plants, roots do not produce monoterpenes, but both exhibit a highly conserved terpene synthase sequence. *TPS*3, *TPS*4, and *TPS*5 were highly expressed in the leaves and flowers, whereas *TPS*1 and *TPS*2 were expressed in the flowers and roots, respectively. Thus, the expression pattern of *TPS* in different organs of *M. chamomilla* was positively correlated with the content of terpenoids in the corresponding organs. The researchers obtained eight TPSs from *M. chamomilla*, of which *TPS*1 produced only an α-bisabolol named *MrBBS*. The expression of α-bisabolol synthase (BBS) in the flowers was 24–58 times of that in the leaf, and 2.5 times higher than that in flower buds at the full flowering stage. *BBS* is localized in cytosol, and the enantiomeric purity of (−)-α-sweet nosafol with a purity of 98% terpenes can be synthesized in vitro [52]. Guo [54] found that the *BBS* gene is located in chloroplasts, and the *BBS*1 protein exhibits α-bisabolol synthase activity, which can catalyze FPP to α-bisabolol. *BBS*1 had the highest homology to the TPS gene of *Artemisia annua*, whereas *BBS*2 had the highest homology to the TPS gene of *M. chamomilla*. The relative expression of the *BBS* gene was highest in the roots and lowest in the stem, which was consistent with the results of Irmisch et al. [53]. *TPS*7 is the corresponding terpene single-product enzyme for the biosynthesis of α-bisabolol, and it can produce α-bisabolol alone with the substrate FPP. No significant correlation was noted between transcription and the corresponding α-bisabolol content in flowers, whereas the correlation between the concentration and the effectiveness of FPP substrates in flowers was strong [55]. Germacrene D synthase (GDS) 1/2/3 demonstrate farnesene synthase, germacrene D synthase, and germacrene A synthase (GAS) activities, respectively, and act on the common substrate FPP to produce EFS. Subcellular localization showed that *GDS*1,

*GDS*2, and *GDS*3 were located in both the cytoplasm and nucleus [56]. The expression of *GDS*1/2/3 in young flowers was higher than that in old flowers; their expression was highly correlated with the content of the final essential oil product, and their overexpression could lead to the specific accumulation of γ-muurolene in the hairy root. Ling et al. [56] also established a transformation efficiency of more than 90% of the *M. chamomilla* hairy root transgenic system, which is suitable for the study of gene function in the root. GAS, as a key enzyme for the synthesis of *M. chamomilla* terpenes, can catalyze the biosynthesis of sesquiterpene, which is a sesquiterpene synthase protein with high homology with the *GAS* gene sequence of other Asteraceae plants. It is widely expressed in various organs and tissues, but it is highly expressed in flowers [57].

The concentration of FPP is one of the key genes in the production of α-bisabolol, and studies have shown that FPP is most important for the production of α- bisabolol in the early stages of terpene biosynthesis [55]. The biosynthesis of sesquiterpene and squalene belongs to the isoprenoid pathway, and they are two important branches of FPP metabolism. Sesquiterpene synthase catalyzes the synthesis of the sesquiterpene skeleton with 1 molecule of FPP, and squalene synthase (SQS) catalyzes two molecules of FPP to synthesize one squalene molecule. The protein secondary structure of the *SQS* gene in chamomile is mainly an α helix (67.97%), which is homologous to *Artemisia annua* and *Arabidopsis thaliana*, and is well conserved in plant molecular evolution. However, the relative expression of the *SQS* gene in different organs is significant; the expression in the tubular corolla is higher than that in the ligulate corolla and stem, the expression in the leaves is higher than that in the tongue-like corolla, and the lowest expression is found in the stem. The *SQS* gene has squalene synthase activity, which can catalyze the synthesis of squalene by FPP, competitively bind to FPP substrates, reduce the binding of sesquiterpene synthase to FPP substrates, and inhibit the synthesis rate of sesquiterpene [58]. FPPS, as a key precursor of sesquiterpene biosynthesis, has a protein sequence that is highly homologous with that of other plants, and the expression of FPPS is higher in the roots than in the other organs and tissues. Overexpression of FPPS can increase the sesquiterpene content, which is an effective way to increase the content of plant active ingredients [59].

As one of the largest families of transcription factors in higher plants, WRKY transcription factors can interact specifically with the W-box cis-acting element. W-box mainly exists in the promoter region of resistance genes related to pest resistance, drought, low temperature, salinity, etc. It regulates plant resistance by mediating hormone signal transduction pathways [60]. A W-box cis-acting element can be found in the promoter sequence of most enzyme genes involved in the terpene biosynthesis of *M. chamomilla*, of which 42 WRKY transcription factors have been identified that may be involved in regulating the terpene synthesis of *M. chamomilla*. *MrWRKY* is commonly expressed in at least one tissue, and the significant difference in the expression levels indicates that *MrWRKY* is organ-specific, which may be related to its specific developmental and metabolic functions in a particular tissue [61]. After treatment with 5 mmol·L$^{-1}$ geraniol and a subculture for 2 h, about 80% of the DNA in the cells of the *M. chamomilla* bud was fragmented, which intensified with time; after 4 h, it was almost completely divided. Geraniol is the most potent apoptosis-inducing agent in terpenes, and the nucleus-concentrated rupture it causes may induce apoptosis of the original cells of *M. chamomilla* buds [62]. Ashida et al. [63] artificially induced two geraniol-inducible response factors, *McEREBP1* and *McWRKY1*, from the adventitious buds of *M. chamomilla*, with the highest expression occurring in geraniol-inducible cells 1 h later. *McEREBP1* may be an activator of GCC box-mediated gene expression induced by geraniol treatment, which induces GCC box-mediated defense genes [63].

Potpourri is a mixture of volatile compounds released from the flower organs of vascular plants, and terpenoids are the main components of potpourri volatiles [64]. Chamazulene and α-bisabolol, the main components of chamomile volatile oil, were positively correlated with the expression of *CPR* and *TPS* genes, and the expression of these two key enzymes and genes can effectively increase the content of *M. chamomilla* oil and α-bisabolol, thereby increasing the medicinal value and economic potential of *M. chamomilla*. According

to the above studies, the synthesis of terpenoids and sesquiterpenoids in *M. chamomilla* is related to the catalysis and expression of *AATC, PMK, MVK, HMGS, HMGR, FPS, βFS, GDS,* and *GAS*. The expression patterns in different organs of *M. chamomilla* were positively correlated with the content of terpenoids in corresponding organs. These genes and enzymes are most expressed in flowers, whereas the *SQS* gene of *M. chamomilla* competitively binds to the FPP substrate, inhibiting the rate of sesquiterpene synthesis. Given that terpenes can be induced by pest erosion and that young tissues are more susceptible to insect attack, the expression of GDS synthase in young flowers is higher than that in old flowers. In addition, the exogenous hormone MeJA can not only upregulate the expression of *McHMGS, MrFPS,* and *βFS* genes but also induce the expression of FPPS and (E)-b acetylene synthase and promote the biosynthesis of sesquiterpenoids (Figure 2). At present, most studies on the gene identification and isolation of terpenoids in *M. chamomilla* have focused on the synthesis of medicinal active ingredients and their economic value. Strengthening research and innovation in the synthesis regulation mechanism of *M. chamomilla* medicinal substances could effectively improve the biological and economic value of *M. chamomilla*, as well as lay the foundation for further research.

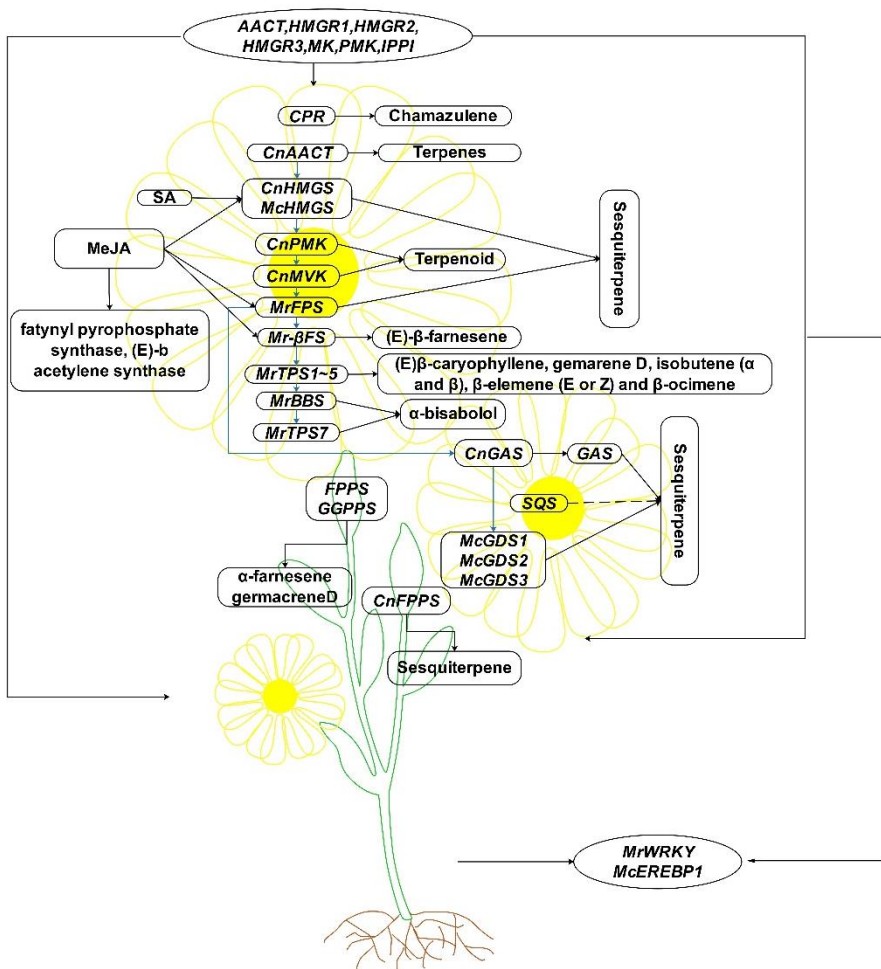

**Figure 2.** Secondary metabolic organs and products of *M. chamomilla*. Enzyme abbreviations: AACT (acetyl-CoA C-acetyltransferase); CAS (germacrene A synthase e); CPR (cytochrome P450 reductase); FPPS (farnesyl pyrophosphate synthase); GGPPS (geranylgeranyl diphosphate synthase); HMGR (3-hydroxy-3-methylglutaryl-CoA reductase); IPPI (isopentenyl diphosphate-isomerase); MeJA (methyl jasmonate); MK (MVA kinase); PMK (phospho-MVA kinase); SA (salicylic acid); SQS (squalene synthase). (The number after the gene is named in the original text, indicating that there are several homologous genes in the gene; for example, *HMGR*1~3 indicates that there are 3 homologous genes in *HMGR*, named *HMGR*1, *HMGR*2, and *HMGR*3).

## 5. Biosynthesis of *M. chamomilla* Flavonoids

Only 0.5% flavonoids have been found in *M. chamomilla*; the content of flavonoids varies, and the content of flavonoids varies between different parts of the flower. The main components of flavonoids are quercetin, luteolin, apigenin, patuletin, coumarin, and marigolin. The matricarin and matricin of guaianolide; the D-glucose, D-galactose, D-galacturonic acid, D-xylose, L-rhamnose, and L-arabinose of polysaccharide mucilage; and the palmitic, linoleic, ascorbic, stearic, oleic, and caprylyl glycolic acid of organic acid are other known components. In addition, the flower contains tridecane, choline, carotene, dandelion sterol, gum, and amaroid [1,10,65,66]. Apigenin has the highest content in *M. chamomilla* flavonoids, but it is mainly in the form of glycosides. It can interfere with the upregulation of leukocyte adhesion and adhesion protein in human endothelial cells and can be used as an anti-inflammatory agent [67–71]. The extract of *M. chamomilla* has inhibitory effects on both poliovirus and herpes [5], and there are significant differences in the bactericidal intensity of aqueous and alcoholic extracts. A previous study confirmed that the aqueous extract of *M. chamomilla* has stronger inhibitory activity against yeast and other fungi [5]. In other plants, many genes related to flavonoid biosynthesis have been identified, but only a few genes related to flavonoid biosynthesis have been reported in *M. chamomilla*. *M. chamomilla* flavanone 3-hydroxylase (F3H) is a family gene, and is one of the early core enzymes in flavonoid biosynthesis, with a typical *F3H* family protein structure and the same amino acid conserved region, sharing an evolutionary ancestor with *F3H* in other plants. The expression of *F3H*s in different tissues significantly differs; *F3H*1, *F3H*2, and *F3H*3 genes was found to be most expressed in the flowers, whereas the *F3H*4 gene was most expressed in the roots. The accumulation of *F3H*s may promote the production of flavonoids in *M. chamomilla* flowers and plays an important role in flavonoid metabolism [72].

## 6. Other Research Advances

### 6.1. Effects of Biological and Abiotic Stress on Essential Oil Composition

Environmental factors, such as temperature, light, photoperiod, nutrition, moisture, and soil salinity, all have a strong influence on the content and quality of essential oil [73,74]. Under normal and thermal stress conditions, the contents of chamazulene and $\alpha$-bisabolol in three kinds of *M. chamomilla* (Bodegold, Bona, and Bushehr) treated with SA significantly increased, but thermal stress did not affect the quality of bisabolol oxide A [75]. The content of $\alpha$-bisabolol is one of the important indicators used to measure the medicinal value of *M. chamomilla*. Spraying a suitable concentration of 5-aminolevulinic acid (ALA) on the leaves of *M. chamomilla* can improve its growth and increase the content of $\alpha$-bisabolol [76]. Salt water irrigation also affects and alters chamomilla's morphology and its agronomic and phytochemical properties. Nitrogen is an essential substance for plant growth and metabolism [77]. Plants must balance the metabolism of carbon-rich metabolites under nitrogen limitation [78]. Phenylalanine ammonia-lyase (PAL) is a key enzyme in the biosynthesis of flavonoid, benzoic acid, and coumarin, and its activity can limit the accumulation of phenolic secondary metabolites. Higher nitrogen concentration in the field may promote plant growth, increase chlorophyll content, and finally, increase the content of phenolic metabolites with antioxidant activity such as umbelliferone in *M. chamomilla* [79].

Heavy metals have a toxic effect on organisms, and the absorption of cadmium and copper will not only reduce the oxidation state of *M. chamomilla* but also affect its water balance and antioxidant capacity [80]. Herniasin is a coumarin-related compound that is unaffected by Cd, but its precursors (Z)- and (E)-2-$\beta$-D-glucopyranosyloxy-4-methoxycinnamic acids (GMCAs) increase significantly [81]. *M. chamomilla* plants selected from areas of medium and high heavy metal levels (Cu, Mn, Ni, Pb, and Zn) show an increase in their contents of low-molecular-weight antioxidants (LMAOs), including flavonoids [82]. Secondary metabolites play an important role in plant defense against biological stress, and the release or accumulation of terpenoids may be affected by herbivores or pathogens. The primary and secondary metabolites of plants undergo major changes when attacked by herbivorous

insects, such as an increase in toxic substances and the production of anti-digestion and anti-nutritional compounds [83]. The invasion of viruses, bacteria, and fungi (*Fusarium*, *Gibberella*, *Puccinia matricariae* SYD, and *Puccinia tanaceti*) directly affects the quality of plants. *M. chamomilla* is one of the host plants of lettuce vein virus and cabbage black ring virus, and damage to its roots, stems, flowers, and leaves affects its essential oil quality [55].

### 6.2. Effect of Geographical Origin on Composition of M. chamomilla

The chemical composition of *M. chamomilla* may vary depending on geographical origin. Raal et al. [84] found that the content of α-bisabolol oxide B was higher than that of α-bisabolol oxide A in *M. chamomilla* from Latvia and Poland, while the opposite was true in *M. chamomilla* from Lithuania and the Netherlands. The contents of artemisinin ketone (7.8%) and (E)-β-farnesene are high in *M. chamomilla* in the United States. The volatile oils of Estonian and Polish *M. chamomilla* have higher contents of chamazulene (3.4–4.9% in total) than those of American samples [84]. The main components of Estonian *M. chamomilla* are α-bisabolol, α-bisabolol oxide A and B, (E)-β-farnesene, chamazulene, and cyclic ether [85]. The main components of *M. chamomilla* are 6-methyl-5-hepten-2-one (0.5–5.4%), artemisyl ketone, artemisol, (E)-β-farnesene (8.6–13.6%), α-bisabolol, α-bisabolol oxide A and B, and the oil of chamazulene obtained via inflorescence [19].

### 6.3. Effect of Seedling Age on Chamomile Composition

There were significant differences in the volatile oil components of *M. chamomilla* from different seedling ages and different regions (Bodegold, Bona, Lutea, Germania, Manzana, Pnos, and Zloty LAN). The contents of sesquiterpene, β-caryophyllene, and β-farnesene were generally dominant in the first 30 days of seedling development in *M. chamomilla*, whereas other terpenes such as β-ocimene, citronella, epoxybutane, and germacrene D varied significantly during seedling growth [55]. α-bisabolol and its oxides were not found during this growth process, which contradicts Mohammad et al. [86]. A large amount of sesquiterpenes that form at the beginning of the development of chamomile seedlings may minimize the production of reactive oxygen species under physiological conditions during seedling growth. *M. chamomilla* in Manzana and Pnos began to produce spiroethers after 30 days, while those in the other five locations continued to accumulate high levels in the first 40 days. In general, a young plant will show stronger defense ability than a mature plant, allowing the plant to adapt to different kinds of stress [87]. Chamomilla seedlings produce significantly lower concentrations of total metabolites in the stems, leaves, buds, or flowers than mature individuals. As the seedlings develop, the production of primary metabolite may be more concentrated upon growth and development, and the production of secondary metabolites is inhibited in the larval stage. In the study of Mondal [55], the complex terpene mixtures produced from the vegetative parts of *M. chamomilla* did not contain α-bisabolol, and there were significant differences in the synthesis of terpene metabolites. The variability of their metabolite production affected all ecological interactions. Regardless of the variety of *M. chamomilla*, α-bisabolol and its oxides are not present in the stems and leaves, and the content of secondary metabolites in young organs is greater than that in mature leaves or stems. This is consistent with the results of Sarrou et al. [88], who investigating qualitative and quantitative differences in the secondary metabolites of *Salvia* plants among populations of same species [55].

### 6.4. Metabolic Differences in Mature Varieties from Different Regions

In general, the number of metabolites varies significantly between germplasms. The main metabolite of the stem and leaf was found to be sesquiterpene, and the concentration of the only monoterpene β-ocimene was significantly different in different germplasm resources. Except Bodegold and Germania, the contents in the leaves of 13 germplasm resources were higher [55]. The regulation of sesquiterpene is independent in the stem and leaf; the stem usually contains high concentrations of α-terpineol, aristolochia, and longifolene, whereas the leaves contain high concentrations of sesquiterpene dicycloguimarene,

α-farnesene, and β-farnesene. The content of sesquiterpenes was highest in the stems of Bona but higher in the leaves of Goral and Lazur. In Bodegold, Camoflora, and Manzana, the contents of β-caryophyllene, β-farnesene, and germacrene D, were higher in the stems than in the leaves. In Bohemia, Camoflora, Lazur, Margaritar, and Zloty *M. chamomilla*, the content of bicyclic germacrene in the stems and leaves varied greatly. In addition to Bona, Germania, Lutea, and Pnos, the content of α-farnesene in the leaves was higher than that in the stems, whereas the difference in ZlotyLan was larger. Except Bodegold, Bohemia, and Goral, spiroether concentrations in the leaves were generally higher. In Lazur, Lutea, and Margaritar, the concentration of spiroether in the stems was significantly lower than that in the leaves. The stem and leaf contents of α-gurjunene, aristolene, and longipinene were significantly different in Bohemia, Goral, Lazur, and Lutea. The differences in inner bicyclic ether among different species are significant; the contents of inner bicyclic ether in the leaves were higher than those in the stems, except for Bodegold, Bohemia, Bona, and Zloty Lan [55].

The volatile oil components in flower buds and flowers are mainly sesquiterpene. Monoterpenes such as β-ocimene and γ-terpinene were found in the tested samples, while sabinene, β-geranene, and α-pinene, in extremely low concentrations, were found in only a few samples. Monoterpenes are usually highest in flowers and vary from individual to individual. Compared with other monoterpenes and sesquiterpenes, spiroethers and dicyclic ethers have higher concentrations in buds and flowers, and higher concentrations in flowers than in buds. Margaritar has the highest content of the two ethers [55]. α-Bisabolol and its oxides differ greatly in their analytical materials; except for Bohemia and Bona, the content of α-bisabolol oxide A in the flowers is mostly higher than that in the buds. No α-bisabolol oxide A was found in the buds of Bona, Camoflora, Germania, Goral, Lazur, or Lutea, whereas a small amount of α-bisabolol oxide B was found in Argenmilla, Goral, and Pohorelicky velkovety. With the exception of Bohemia, the amount of α-bisabolol in flowers was higher than that in flower buds [55].

Research Perspectives: Known as the "star of herbal medicine," *M. chamomilla* is not only one of the top five best-selling herbs in the world but also an important additive in food, medicine, and cosmetics. As a stress-resistant medicinal and aromatic plant, it has great development value and economic value. Compared with Western countries, the interest in and research on *M. chamomilla* in China started late, and most of the domestic and foreign studies focus on its pharmacological effects. Although several terpenes have been isolated and identified from *M. chamomilla*, only a few studies reported its flavonoids and other compounds. Future work should strengthen the research and innovation of the synthetic regulation mechanism of *M. chamomilla*, as well as providing a theoretical basis for improving its medicinal value and economic value.

**Author Contributions:** H.W.: original draft preparation, review, and editing the manuscript. J.Y. and F.X.: review and editing the manuscript. K.Y. and L.D.: proofreading and formatting. All authors have read and agreed to the published version of the manuscript.

**Funding:** This work was supported by the National Natural Science Foundation of China (grant number 31400603).

**Institutional Review Board Statement:** Not applicable.

**Informed Consent Statement:** Not applicable.

**Data Availability Statement:** Not applicable.

**Conflicts of Interest:** The authors declare no conflict of interest.

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
