# Peer review of "Classification, Distribution, Biosynthesis, and Regulation of Secondary Metabolites in Matricaria chamomilla"

_horticulturae, doi:10.3390/horticulturae8121135_

Round 1
Reviewer 1 Report
The current article entitled “Classification, distribution, biosynthesis, and regulation of secondary metabolites in Matricaria chamomilla” By Wu et al., summarizes the studies on the secondary metabolites of medicinally active components in M. chamomilla, for their biosynthesis pathways and regulation. The whole manuscript needs some grammatical corrections. However, some major issues need to be addressed before taking any decision.
1. Introduction
1. “. It is also used for pain relief and sedation. In recent years, many studies have examined the medicinal active ingredients, pharmacological efficacy, plant physiology, and other aspects of M. chamomilla. Here, we summarize the studies on the secondary metabolites of medicinal” used medicinally instead of medicinal.
2. Chamaemelum nobile, derived from the family Asteraceae, and M. chamomilla” also mention the family for chamomilla and reframe the sentence.
3. “Given that the two species have very similar appearances, and the standard of medicinal materials is low, C. nobile is often mistaken for M. chamomilla during medicinal material acquisition.” Rewrite it.
4. “on the volatile oil of M. chamomilla, M. chamomilla is a medicinal herb with antiseptic, anti-inflammatory, antispasmodic, and soothing” rewrite.
5. Write the objectives to write this review article.
6. In biological characteristics, paragraph 5 lines start with the same article “the”. Change it.
7. “M. chamomilla is a year-old dwarf herb”… meaning of the sentence is not clear. Please rewrite.
8. In biological characteristics paragraph, 1 come after paragraph 2 because first describe the habitat and after that characteristic of the plant.
9. Remove the space between 30 ⁰C. And use the same format in the whole manuscript.
10. “including 28 kinds of terpenoids, 36 kinds of flavonoids, and 56 kinds of organic acids and other compounds such as coumarins” first describe the class with the maximum number of compounds.
11. “Volatile oils, as the main source of medicinal active ingredients of M. chamomilla, are terpenoid compounds, and monoterpenes and sesquiterpenes are the most abundant, of which oxygenated derivatives have less content but are mostly aromatic”. Rewrite these lines.
12. “:It is a sesquiterpene lactone, the product of the reaction of chamazulene during distillation and has significant anti-inflammatory, antispasmodic, antibacterial, and bacteriostatic activities, free radical scavenging”. Rewrite the lines
13. Alphabetically arranged the abbreviations in the figure legends
14. Explain the figures in the figure legend.
15. The abbreviation of “Farnesyl diphosphate synthase, FPPS” used FDPS. Because pp is used for pyrophosphatase
16. “salinity and so on, which and regulate plant resistance by mediating hormone signal transduction pathways” rewrite the line.
17. “Potpourri is a mixture of volatile compounds released from the flower organs of vascular plant, and terpenoids are the main components of potpourri volatiles”. Why does that paragraph not contain even a single citation?
18. “it is widely used in food, beverages, tobacco, daily chemicals, medicine, and other fields, with good market development potential of new agricultural plants” used “for”
19. “Previous study have found that M. chamomilla contains more than 120 kinds of medicinal active ingredients” the previous studies
20. “The expression of genes involved in the mevalonic acid (MVA) pathway are higher in leaves” is
21. “Chamazulene and α-bisabolol, the main components of chamomile volatile oil, were positively correlated with the expression of CPR and TPS gene” change it to genes.
22. “Only 0.5% flavonoid was found in M. chamomilla; the content of flavonoid varies, and the content of flavonoid varies” flavonoids
23. “Phenylalanine ammonia lyase (PAL) is a key enzyme in the biosynthesis of flavonoid, benzoic acid, and coumarin, and its activity can limit the accumulation of phenolic secondary metabolites” add hyphen
24. “The main components of M. chamomilla ar 6-methyl-5-hepten-2-one (0.5%–5.4%),”
25. “This growth process did not include α-bisabolol and its oxides, which was contradicted by Mohammad et al [93].” Include
2. General comments:
The whole manuscript contains grammatical and formatting mistakes. So remove them.
The plant name should be in italic.
Author Response
Dear Reviewer 1:
Thank you for your letter and the reviewers’ comments concerning our manuscript entitled “Classification, distribution, biosynthesis, and regulation of secondary metabolites in Matricaria chamomilla” (Manuscript ID: horticulturae-2008817).
Those comments are all valuable and very helpful for revising and improving our paper, as well as the important guiding significance to our work. We have studied comments carefully and have made correction which we hope meet with approval. Revised portion are marked in red in the paper. The main corrections in the paper and the responds to the reviewer’s comments are as following:
- Introduction
“. It is also used for pain relief and sedation. In recent years, many studies have examined the medicinal active ingredients, pharmacological efficacy, plant physiology, and other aspects of M. chamomilla. Here, we summarize the studies on the secondary metabolites of medicinal” used medicinally instead of medicinal.
Response: I'm sorry that we made such a mistake. According to the reviewer's suggestion, we have changed “medicinal” to “medicinally”.
- Chamaemelum nobile, derived from the family Asteraceae, and chamomilla”also mention the family for chamomilla and reframe the sentence.
Response: I'm sorry that due to our negligence, the meaning of this sentence is not accurate. It should be changed to “There are two species of chamomile commonly used: Chamaemelum nobile and M. chamomilla, derived from the family Asteraceae[1-2].”
- “Given that the two species have very similar appearances, and the standard of medicinal materials is low, nobile is often mistaken for M. chamomilla during medicinal material acquisition.” Rewrite it.
Response: I'm sorry that due to our negligence, the meaning of this sentence is not accurate. It should be changed to “Because the appearance of the two is very similar and the standard of medicinal materials is low, the C. nobile is often mistaken for the M. chamomilla in the process of purchasing medicinal materials.”
- “on the volatile oil of chamomilla, M. chamomillais a medicinal herb with antiseptic, anti-inflammatory, antispasmodic, and soothing” rewrite.
Response: I'm sorry that due to our negligence, the meaning of this sentence is not accurate. It should be changed to “According to a large number of pharmacological studies on the volatile oil of M. chamomilla, the plant is a medicinal herb with antiseptic, anti-inflammatory, antispasmodic, and soothing effects [3].”
- Write the objectives to write this review article.
Response: We are very grateful for your providing of the suggestions above. According to the reviewer's good instruction, we defined the purpose of the review more clearly in this paper: “This paper reviews the types of terpenoids and flavonoid in M. chamomilla, the synthesis pathways, regulatory factors and functions of secondary metabolites, and the identification, regulatory mechanisms and effects of biosynthesis-related genes. We also explored the effects of exogenous hormones, biotic and abiotic stress, geographical environment and other factors on the secondary metabolites and content of M. chamomilla. Hope to lay a foundation for further research and utilization of M. chamomilla.”
- In biological characteristics, paragraph 5 lines start with the same article “the”. Change it.
Response: Corrected accordingly.
- “M. chamomillais a year-old dwarf herb”… meaning of the sentence is not clear. Please rewrite.
Response: It should be changed to “M. chamomilla is an annual herb, about 30–40cm, with a straight, smooth, hairless stem and many branches.”
- In biological characteristics paragraph, 1 come after paragraph 2 because first describe the habitat and after that characteristic of the plant.
Response: Corrected accordingly.
- Remove the space between 30 ⁰C. And use the same format in the whole manuscript.
Response: Corrected accordingly.
- “including 28 kinds of terpenoids, 36 kinds of flavonoids, and 56 kinds of organic acids and other compounds such as coumarins” first describe the class with the maximum number of compounds.
Response: At your suggestion, we have changed it to “The previous studies have found that M. chamomilla contains more than 120 types of medicinal active ingredients, including 56 types of organic acids 36 types of flavonoids, 28 types of terpenoidsand other compounds such as coumarins [5].”
- “Volatile oils, as the main source of medicinal active ingredients of M. chamomilla, are terpenoid compounds, and monoterpenes and sesquiterpenes are the most abundant, of which oxygenated derivatives have less content but are mostly aromatic”. Rewrite these lines.
Response: I'm sorry that due to our negligence, the meaning of this sentence is not accurate. It should be changed to “As the main medicinal active ingredient source of volatile oil, the important substances are terpenoids. Monoterpenes and sesquiterpenes are the most abundant, of which oxygen-containing derivatives are less abundant but mostly have aroma.”
- “:It is a sesquiterpene lactone, the product of the reaction of chamazulene during distillation and has significant anti-inflammatory, antispasmodic, antibacterial, and bacteriostatic activities, free radical scavenging”. Rewrite the lines
Response: We changed it to “Chamazulene is a sesquiterpene lactone, which is the product of the reaction of pre-chamazulene during distillation. It has significant anti-inflammatory, antispasmodic, antibacterial, and bacteriostatic activities, and can also scavenge free radicals [11,26,27].”
- Alphabetically arranged the abbreviations in the figure legends
Response: According to your suggestion, we arrange the figure legends in alphabetical order, as follows “Figure 1. Terpenoid synthesis pathway in M. chamomilla. Enzyme abbreviations: AACT (Acetyl-CoA C-acetyltransferase); DMAPP (Dimethylallyl diphosphate); DXP (1-Deoxy-xylose-5-phosphate); DXS (1-Deoxy-D-xylulose 5-phosphate synthase); FPP (Farnesyl pyrophosphate); FPPS (Farnesyl pyrophosphate synthase); GGAPP (Geranylgeranyl pyrophosphate); GPP (Geranyl diphosphate); HMGR (3-Hydroxy-3-methylglutaryl-CoA reductase); HMGS (3-Hydroxy-3-methylglutaryl-CoA synthase); IPP (Isoprene diphosphate); MEP (2-C-methyl-D-erythritol 4-phosphate); MK (MVA kinase); MPDC (Diphospho-MVA decarboxylase); MVA (Mevalonate); MVAP (mevalonate-5-pyrophosphate); MVAPP (Mevalonate 5-diphosphate); PMK (Phosphomevalonate kinase); SQ(squalene); SQS (Squalene synthase); TPS (Terpene synthase).
Figure 2. Secondary metabolic organs and products of M. chamomilla. Enzyme abbreviations:AACT (Acetyl-CoA C-acetyltransferase); CAS (Germacrene A synthase e); CPR (Cytochrome P450 reductase); FPPS (Farnesyl pyrophosphate synthase); GGPPS (Geranylgeranyl diphosphate synthase); HMGR (3-Hydroxy-3-methylglutaryl-CoA reductase); IPPI (Isopentenyl diphosphate-isomerase); MeJA (Methyl Jasmonate); MK (MVA kinase); PMK (Phospho-MVA kinase); SA (Salicylic acid); SQS (Squalene synthase).”
- Explain the figures in the figure legend.
Response: Corrected accordingly.
- The abbreviation of “Farnesyl diphosphate synthase, FPPS” used FDPS. Because pp is used for pyrophosphatase
Response: As per your suggestion, the abbreviation of the related gene has been revised
- “salinity and so on, whichand regulate plant resistance by mediating hormone signal transduction pathways” rewrite the line.
Response: We changed it to “W-box mainly exists in the promoter region of resistance genes related to pest resistance, drought, low temperature, salinity and so on. It regulates plant resistance by mediating hormone signal transduction pathways [60].”
- “Potpourri is a mixture of volatile compounds released from the flower organs of vascular plant, and terpenoids are the main components of potpourri volatiles”. Why does that paragraph not contain even a single citation?
Response: We are very grateful for your providing of the suggestions above. According to the reviewer's good instruction, we have added the reference in this paper.
- “it is widely used in food, beverages, tobacco, daily chemicals, medicine, and other fields, with good market development potential of new agricultural plants” used “for”
Response: Corrected accordingly.
- “Previous study has found that chamomillacontains more than 120 kinds of medicinal active ingredients” the previous studies
Response: Corrected accordingly.
- “The expression of genes involved in the mevalonic acid (MVA) pathway are higher in leaves” is
Response: Corrected accordingly.
- “Chamazulene and α-bisabolol, the main components of chamomile volatile oil, were positively correlated with the expression of CPRand TPS gene” change it to genes.
Response: Corrected accordingly.
- “Only 0.5% flavonoid was found in chamomilla; the content of flavonoid varies, and the content of flavonoid varies” flavonoids
Response: Corrected accordingly.
- “Phenylalanine ammonia lyase (PAL) is a key enzyme in the biosynthesis of flavonoid, benzoic acid, and coumarin, and its activity can limit the accumulation of phenolic secondary metabolites” add hyphen
Response: Corrected accordingly.
- “The main components of chamomillaar 6-methyl-5-hepten-2-one (0.5%–5.4%),”
Response: Corrected accordingly.
- “This growth process did not include α-bisabolol and its oxides, which was contradicted by Mohammad et al [93].” Include
Response: We changed it to “No α-bisabolol and its oxides were found during this growth process, which contradicts Mohammad et al [86].”
- General comments:
- The whole manuscript contains grammatical and formatting mistakes. So remove them.
Response: I'm sorry that we made such a mistake. According to the reviewer's suggestion, we carefully examined the article and revised its grammar and spelling mistakes.
- The plant name should be in italic.
Response: I'm sorry that we made such a mistake. According to the reviewer's suggestion, we changed all the plant names to italics.
Reviewer 2 Report
Please complete the manuscript by newer references.
Author Response
Dear Reviewer 2,
Thank you for your letter and the reviewers’ comments concerning our manuscript entitled “Classification, distribution, biosynthesis, and regulation of secondary metabolites in Matricaria chamomilla” (Manuscript ID: horticulturae-2008817).
Those comments are all valuable and very helpful for revising and improving our paper, as well as the important guiding significance to our work. We have studied comments carefully and have made correction which we hope meet with approval. Revised portion are marked in red in the paper. The main corrections in the paper and the responds to the reviewer’s comments are as following:
- English language and style are fine/minor spell check required
Response: I'm sorry that we made such a mistake. According to the reviewer's suggestion, we carefully examined the article and revised its grammar and spelling mistakes.
- Please complete the manuscript by newer references.
Response: We are very grateful for your providing of the suggestions above. According to the reviewer's good instruction, we have updated the references references in this paper. The modifications are as follows:
References
- Liu X.M.; Meng X.X.; Zhang W.W.; Liao Y.L.; Chang J.; Xu F. Tissue Culture Technology of Matricaria chamomilla North. Hortic. 2018, 2, 72-76.
- Han S.L.; Li X.X.; Mian Q.H.; Lan W.; Liu Y. Comparison of antioxidant activity between two species of chamomiles produced in Xinjiang by TLC-bioautography. China J. Chin. Mater. Med. 2013, 38, 193-198.
- Wan W.T.; Song Y.J.; Xu L.J.; Xiao P.G.; Miao J.H. An overview on modern research and application potency of Chamomile. Chin. Med. 2019, 21, 260-265.
- Bhattacharjee S.K. Handbook of aromatic plants. India: Pointer Publishers, 2005; pp. 277–279.
- Xia Q.X.; Bai H.T.; Sun L.C.; Gao T.G.; Jiang C.D.; Shi L. Research progress on active composition and practical application of medicinal plants of Matricaria recutita. Acta Hortic. Sinica 2012, 39, 1859-1864.
- Shiva M.P.; Lehri A.; Shiva A. Matricaria-Chamomilla/IAromatic and medicinal plants. International Book Distributors, India, 2002; pp. 223–228.
- Xu L. United States Pharmacopeia (24th Edition): Chamomile. World Notes Plant Med. 2002, 2, 82-83.
- Wu Z.Y. Flora of China. Beijing:Science Press 2010; pp. 49–50.
- Yang Y.S.; Pan L.S. Isolation and structural determination of flavones from Matricaria chamomilla Appl. Chem. Ind. 2008, 37, 697–698.
- Zhou B.T.; Li X.Z. Study of the chemical composition of chamomile. Hunan Coll. Tradit. Chin. Med. 2001, 21, 27–28.
- Zhao Y.F. Chemical constituents and quality standard of Matricaria chamomilla as uygur medicine. MA thesis, China Academy of Chinese Medical Sciences, Beijing, China, 2018.
- Krüger H. Characterisation of Chamomile Volatiles by Simultaneous Distillation Solid-Phase Extraction in Comparison to Hydrodistillation and Simultaneous Distillation Extraction. Planta Med. 2010, 76, 843–846.
- Schilcher H. Die Kamille: Handbuch für Ärzte,Apotheker und andere Naturwissenschaftler. Wissenschaftliche Verlagsgesellschaft, Stuttgart: Wiss. Verl, 1987; pp. 59–61.
- Kazemi M. Chemical Composition and Antimicrobial Activity of Essential Oil of Matricaria recutita. J. Food Prop. 2015, 18, 1784-1792.
- Wesolowska A.; Grzeszczuk M.; Kulpa D. Propagation Method and Distillation Apparatus Type Affect Essential Oil from Different Parts of Matricaria recutita Plants. J. Essent. Oil Bear. Pl. 2015, 18:179-194.
- Zhao Y.F.; Zhang D.; Liang X.X.; Yang L.X.; Sun P.; Ma Y.; Wang K.; Chang X.Q.; Yang L. Chemical constituents from Matricaria chamomilla(Ⅰ). J. Chin. Pharm. Sci. 2018, 27, 324-331.
- Cordero C.; Sgorbini B.; Rubiolo P.; Belliardo F.; Liberto E.; Bicchi C. Headspace–solid‐phase microextraction fast GC in combination with principal component analysis as a tool to classify different chemotypes of chamomile flower‐heads (Matricaria Recutita). Phytochem. Analysis 2006, 17, 217-225.
- Xu Y.B.; Tang H.; Zhu S.Y.; Zhe W.; Wang K.; Mao D.S.; Fu L.; Chen R.R. Analysis of Volatile Components of Chamomile Oil of Different Origins by Gas Chromatography Time-of-Flight Mass Spectrometry. Technol. Food Ind. 2015, 36, 6.
- Das M.; Ram G.; Singh A.; Mallavarapu G.R.; Ramesh S.; Ram M.; Kumar S. Volatile constituents of different plant parts of Chamomilla recutita Rausch grown in the Indo-Gangetic plains. Flavour. Fragr. J. 2002, 17, 9–12.
- Povh N.P.; Garcia C.A.; Marques M.O.M.; Meireles M.A.A. Extraction of essential oil and oleoresin from chamomile (Chamomila recutita [L.] Rauschert) by steam distillation and extraction with organic solvents: a process design approach. Bras. Pl. Med. 2001, 4, 1–8.
- McKay D.L.; Blimiberg J.B. A review of the bioactivity and potential health benefits of chamomile tea (Matricaria recutita). Phytother. Res. 2006, 20, 519–530.
- Russell K.; Jacob S.E. Bisabolol. Dermatitis 2010, 21, 57–58.
- Uno M.; Kokuryo T.; Yokoyama Y.; Senga T.; Nagino M. α-Bisabolol Inhibits Invasiveness and Motility in Pancreatic Cancer Through KISS1R Activation. Anticancer 2016, 36, 583–589.
- Yan H.B.; Xu R.X. Effect of α-bisabolol on migration and invasion of glioblastoma cells. J. Chin. Pla Med. Sch. 2018, 39, 699–706.
- Sharkey T.D.; Gray D.W.; Pell H.K.; Breneman S.R.; Topper L. Isoprene synthase genes form a monophyletic clade of acyclic terpene synthases in the TPS-B terpene synthase family. Int. J. Org. Evol. 2013, 67, 1026–1040.
- Schilcher H.; Imming P.; Goeters S. Active Chemical Constituents of Matricaria chamomilla syn. Chamomilla recutita (L.) Rauschert. Chamomile ind. profile. 2005, 1, 56–76.
- Farhoudi R. Chemical constituents and antioxidant properties of Matricaria Recutita and Chamaemelum nobile essential oil growing wild in the south west of Iran. Essent. Oil Bear. Pl. 2013, 16, 531–537.
- Schnee C.; Köllner T.G.; Held M.; Turlings T.C.J.; Gershenzon J.; Degenhardt J. The products of a single maize sesquiterpene synthase form a volatile defense signal that attracts natural enemies of maize herbivores. Natl. A. Sci. 2006, 103, 1129–1134.
- Köllner T.G.; Held M.; Lenk C.; Hiltpold I.; Turlings T.C.; Gershenzon J.; Degenhardt J. A Maize (E)-β-Caryophyllene Synthase Implicated in Indirect Defense Responses against Herbivores Is Not Expressed in Most American Maize Plant Cell 2008, 20, 482–494.
- Yu L.T.; Cheng C.L.; Cheng X.W.; Huan H.W.; Ling S.; Lin Y.; Wei J.; Xiao R.Y.; Lu J.Z.; Zhan F.; Chun L.; Yi Y. Analysis of terpenoid biosynthesis pathways in german chamomile (Matricaria recutita) and roman chamomile (Chamaemelum nobile) based on co-expression networks. Genomics 2020, 112, 1055-1064.
- Wright G.A.; Schiestl F.P. The evolution of floral scent: the influence of olfactory learning by insect pollinators on the honest signalling of floral rewards. Ecol. 2009, 23, 841–851.
- Zhang W.W.; Tao T.T.; Liu X.M.; Xu F.; Chang J.; Liao Y.L. De novo assembly and comparative transcriptome analysis: novel insights into sesquiterpenoid biosynthesis in Matricaria chamomilla Acta Physiol. Plant. 2018, 40, 1–14.
- Han X.; Wang Y.D.; Chen Y.C.; Lin L. Y.; Wu Q.K.; Christian S. Transcriptome Sequencing and Expression Analysis of Terpenoid Biosynthesis Genes in Litsea cubeba. PloS One 2013, 8: e76890.
- Zhao Y.J.; Chen X.; Zhang M.; Ping S.; Liu Y.J.; Tong Y.R.; Wang X.J.; Huang L.Q.; Gao W. Molecular cloning and characterisation of farnesyl pyrophosphate synthase from Tripterygium wilfordii. PloS One 2015, 10, e0125415.
- Ortiz de Montellano P.R. Acetylenes: cytochrome P450 oxidation and mechanism-based enzyme inactivation. Drug Metab. Rev. 2019, 51, 162–177.
- Ling S.P.; Zhang H.M.; Su S.S.; Zhang X.S.; Liu X.Y.; Pan G.F.; Yuan Y. Molecular Cloning and characterization of a Cytochrome P450 reductase gene (CPR) full-length in Matricaria recutita. Agr. Biotechnol. 2014, 22, 580–589.
- Fu M.; Liu X.; Meng X.; Wang L.; Tan J.; Zhou X.; Xu F. Cloning and Sequence Analysis of an Acetyl-CoA C-Acetyltransferase Gene (AACT) from Chamaemelum nobile. J. Curr. Res. Biosci. Plant Biology 2017, 4, 31–37.
- Yan J.; Meng; Xu F.; Chang J. Molecular cloning and sequence analysis of a phosphomevalonate kinase gene (CnPMK) from Chamaemelum nobile. Int. J. Curr. Res. Biosci. Plant Biology 2016, 3,157–162.
- Meng X.X.; Zhang W.; Xu F.; Yan J.; Liu X.; Liao Y.L.; Chang J. Cloning and sequence analysis of mevalonate kinase gene (CnMVK) from Chamaemelum nobile. J. Curr. Res. Biosci. Plant Biology 2016, 3, 23–28.
- Cheng S.Y.; Wang X.H.; Xu F.; Chen Q.W.; Tao T.T.; Lei J.; Zhang W.W.; Liao Y.L.; Chang J.; Li X.X.; Gulder T.A.M. Cloning, Expression Profiling and Functional Analysis of CnHMGS, a Gene Encoding 3-hydroxy-3-Methylglutaryl Coenzyme A Synthase from Chamaemelum nobile. Molecules 2016, 21, 316.
- SuS.; Liu X.Y.; Pan G.F.; Hou X.J.; Zhang H.M.; Yuan Y. In vitro characterization of a (E)-β-farnesene synthase from Matricaria recutita L. and its up-regulation by methyl jasmonate. Gene 2015, 571, 58–64.
- Tao T.T.; Chen Q.W.; Meng X.X.; Yan J.P.; Xu F.; Chang J. Molecular cloning, characterization, and functional analysis of a gene encoding 3-hydroxy-3-methylglutaryl-coenzyme A synthase from Matricaria chamomilla. Genes Genom. 2016, 38, 1179–1187.
- Su S.S.; Zhang H.M.; Liu X.Y.; Pan G.F.; Ling S.P.; Zhang X.S.; Yang X.M.; Tai Y.L.; Yuan Y. Cloning and characterization of a farnesyl pyrophosphate synthase from Matricaria recutita and its upregulation by methyl jasmonate. Genet. Mol. Res. 2015, 14, 349–361.
- Meng X.X.; Yan J.P.; Liao Y.L.; Chang J.; Xu F. Cloning and expression analysis of HMGR gene from Chamaemelum nobile. Acta Agr. Boreal. Sinica 2016, 31, 68–75.
- Sun J.M. Omics-based study on genes related to volatile terpenoids from Matricaria chamomilla MA thesis. Anhui Agricultural University, Anhui, China, 2018.
- Tai Y.L. Molecular Cloning, Expression of FPS and analysis of salt stress in Matrucarua chamomilla MA thesis, Anhui Agricultural University, Anhui, China, 2012.
- Yang X.M. Study on Rapid Propagation of Matricaria Chamomile L and genetic transformation of FPS gene in Tobacco. MA thesis, Anhui Agricultural University, Anhui, China, 2013.
- Gupta P.; Akhtar N.; Tewari S.K.; Sangwan R.S.; Trivedi P.K. Differential expression of farnesyl diphosphate synthase gene from Withania somnifera in different chemotypes and in response to elicitors. Plant Growth Regul. 2011, 65, 93–100.
- Lan J.B.; Yu R.C.; Yu Y.Y.; Fan Y.P. Molecular cloning and expression of Hedychium coronarium farnesyl pyrophosphate synthase gene and its possible involvement in the biosynthesis of floral and wounding/herbivory induced leaf volatile sesquiterpenoids. Gene 2013, 518, 360–367.
- Xiang L.; Zhao K.; Chen L.Q. Molecular cloning and expression of Chimonanthus praecox farnesyl pyrophosphate synthase gene and its possible involvement in the biosynthesis of floral volatile sesquiterpenoids. Plant Physiol. Bioch. 2010, 48, 845–850.
- Su S.S. Cloning and characterization of (E)-β-farnesene synthase from Matricaria recutita MA thesis, Anhui Agricultural University, Anhui, China, 2015.
- Son Y.J.; Kwon M.; Ro D.K.; Kim S.U. Enantioselective microbial synthesis of the indigenous natural product (−)-α-bisabolol by a sesquiterpene synthase from chamomile (Matricaria recutita). J. 2014, 463, 239–248.
- Irmisch S.; Krause S.T.; Kunert G.; Gershenzon J.; Degenhardt J.; Köllner T.G. The organ-specific expression of terpene synthase genes contributes to the terpene hydrocarbon composition of chamomile essential oils. BMC Plant Biol. 2012, 12, 1–13.
- Guo C.X. Cloning and characterization of α-bisabolol synthase gene (MrBBS) from Matricaria chamomilla MA thesis, Anhui Agricultural University, Anhui, China, 2018.
- Mondal P. Biosynthesis and regulation of terpene production in accessions of chamomile (Matricaria recutita). Ph D thesis, Martin-Luther-Universität Halle-Wittenberg, Halle, Germany, 2020.
- Ling C.C.; Zheng L.J.; Yu X.R.; Wang H.H.; Wang C.X.; Wu H.Y.; Zhang J.; Yao P.; Tai Y.L.; Yuan Y. Cloning and functional analysis of three aphid alarm pheromone genes from German chamomile (Matricaria chamomilla). Plant Sci. 2020, 294, 110463.
- Yan J.P.; Meng X.X.; Zhu L.; Zhang W.W.; Chang J.; Xu F. Molecular cloning and expression analysis of germacrene A synthase gene in Chamaemelum nobile. Tradit. Herb. Drugs 2017, 48, 1851–1859.
- Ling S.P. Molecular Cloning and characterization of Squalene synthase gene and analysis of essential oil from Different organs of Matricaria recutita. MA thesis, Anhui Agricultural University, Anhui, China, 2014.
- Liu X.M.; Tao T.T.; Meng X.X.; Zhang W.W.; Chang J.; Xu F. Cloning and expression analysis of a farnesyl diphosphate synthase (FPPS) gene from Chamaemelum nobile. Bot. Horti Agrobo. 2017, 45, 358–364.
- Zhang T.; Li Z.Q.; Wu G.Q. Role of WRKY Transcription Factor in Plant Response to Stresses. Bull. 2021, 37, 203-215.
- Xiang L.; Zhang W.W.; Qu J.W.; Han H.; Xu F.; Liao Y.L. Identification and Expression Analysis of the WRKY Transcription Factor in Matricaria recutita Mol. Plant Breeding 2020, 18, 2127–2137.
- Izumi S.; Takashima O.; Hirata T. Geraniol Is a Potent Inducer of Apoptosis-like Cell Death in the Cultured Shoot Primordia of Matricaria chamomilla. Bioph. Res. Co. 1999, 259, 519–522.
- Ashida Y.; Nishimoto M.; Matsushima A.; Watanabe J.; Hirata T. Molecular Cloning and mRNA Expression of Geraniol-inducible Genes in Cultured Shoot Primordia of Matricaria chamomilla. Biotech. Bioch. 2002, 66, 2511–2514.
- Li Y.Y. The main composition and affecting factors of aroma volatiles in flowers. Northern Horticult. 2012, 6, 184-189.
- Sharifi-Rad M; Nazaruk J; Polito L; Morais-Braga M.F.B.; Rocha J.E.; Coutinho H.D.M.; Salehi B.; Tabanelli G.; Montanari C.; Del Mar Contreras M.; Yousaf Z.; Setzer W.N.; Verma D.R.; Martorell M.; Sureda A.; Sharifi-Rad J. Matricaria genus as a source of antimicrobial agents: From farm to pharmacy and food applications. Res. 2018, 215, 76-88.
- Sagi S.; Avula B.; Wang Y.H.; Zhao J.; Khan I.A. Quantitative determination of seven chemical constituents and chemo-type differentiation of chamomiles using high-performance thin-layer chromatography. J. Sep. Sci. 2014. 37, 2797-804.
- Mobasheri L.; Khorashadizadeh M.; Safarpour H.; Mohammadi M.; Anani Sarab G.; Askari V.R. Anti-inflammatory activity of ferula assafoetida Oleo-Gum-Resin (Asafoetida) against TNF-α-stimulated Human Umbilical Vein Endothelial Cells (HUVECs). Inflamm. 2022. 31, 5171525.
- Avallone R.; Zanoli P.; Puia G.; Kleinschnitz M.; Schreier P.; Baraldi M. Pharmacological profile of apigenin, a flavonoid isolated from Matricaria chamomilla. Pharmacol. 2000, 59, 1387–1394.
- Švehlíková V.; Bennett R.N.; Mellon F.A.; Needs P.W.; Piacente S.; Kroon P.A.; Bao Y. Isolation, identification and stability of acylated derivatives of apigenin 7-O-glucoside from chamomile (Chamomilla recutita [L.] Rauschert). Phytochemistry 2004, 65, 2323–2332.
- Babenko N.A, Shakhova E.G. Effects of Chamomilla recutita flavonoids on age-related liver sphingolipid turnover in rats. Gerontol. 2006, 41, 32–39.
- Weber B.; Herrmann M.; Hartmann B.; Joppe H.; Schmidt C.O.; Bertram H.J. HPLC/MS and HPLC/NMR as hyphenated techniques for accelerated characterization of the main constituents in Chamomile (Chamomilla recutita [L.] Rauschert). Food Res. Technol. 2008, 226, 755–760.
- Cheng S.Y.; Song Q.; Tian Y.; Liu X.; Wang L.; Mao D.; Zhang W.; Xu F. Characterization and expression analysis of four members genes of flavanone 3-hydroxylase families from chamaemelum nobile. Bot. Horti Agrobo. 2020, 48, 102–115.
- Badi H.N.; Yazdani D.; Ali S.M.; Nazari F. Effects of spacing and harvesting time on herbage yield and quality/quantity of oil in thyme, Thymus vulgaris L. Crop. Prod. 2004, 19, 231–236.
- Banchio E.; Zygadlo J.; Valladares G. R. Quantitative variations in the essential oil of Minthostachys mollis (kunth.) griseb. in response to insects with different feeding habits. Agr. Food Chem. 2005, 53, 6903–6906.
- Ghasemi M.; Babaeian J. N.; Modarresi M.; Bagheri N.; Jamali A. Increase of Chamazulene and α-Bisabolol Contents of the Essential Oil of German Chamomile (Matricaria chamomilla) Using Salicylic Acid Treatments under Normal and Heat Stress Conditions. Foods 2016, 5, 56.
- Liu X.; Zhu L.; Song Q.; Chang J.; Ye J.; Zhang W.; Liao Y.; Xu F. Effects of 5-aminolevulinic Acid on the Photosynthesis, Antioxidant System, and α-Bisabolol Content of Matricaria recutita. Bot. Horti Agrobo. 2018, 46, 418–425.
- Kusano M.; Fukushima A.; Redestig H.; Saito K. Metabolomic approaches toward understanding nitrogen metabolism in plants. Exp. Bot. 2011, 62, 1439–1453.
- Matt P.; Krapp A.; Haake V.; Mock H.P.; Stitt M. Decreased Rubisco activity leads to dramatic changes of nitrate metabolism, amino acid metabolism and the levels of phenylpropanoids and nicotine in tobacco antisense RBCS transformants. Plant J. 2002, 30, 663–677.
- Repčák M.; Imrich J.; Franeková M. Umbelliferone, a stress metabolite of Chamomilla recutita (L.) Rauschert. Plant Physiol. 2001, 158, 1085–1087.
- Maksymiec W. Signaling responses in plants to heavy metal stress. Acta Physiol. Plant. 2007, 29, 177–187.
- Kováčik J.; Tomko J.; Bačkor M.; Repčák M. Matricaria chamomilla is not a hyperaccumulator, but tolerant to cadmium stress. Plant Growth Regul. 2006, 50, 239–247.
- Prokop’ev I.A.; Filippova G.V.; Shein A.A.; Gabyshev D.V. Impact of urban anthropogenic pollution on seed production, morphological and biochemical characteristics of chamomile, Matricaria chamomila Russ. J. Ecol. 2014, 45, 18–23.
- Kessler A.; Baldwin I.T. Plant responses to insect herbivory: the emerging molecular analysis. Rev. Plant Biol. 2002, 53, 299–328.
- Raal A.; Kaur H.; Orav A.; Arak E.; Kailas T.; Muurisepp M. Content and composition of essential oils in some Asteraceae species. Natl A. Sci. 2011, 60, 55–63.
- Orav A.; Kailas T.; Ivask K. Volatile constituents of Matricaria recutita from Estonia. Est. A. Sci. Chem. 2001, 50, 39–45.
- Mohammad R.; Hamid S.; An A.; Norbert D.K.; Patrick V.D. Effects of planting date and seedling age on agro-morphological characteristics, essential oil content and composition of German chamomile (Matricaria chamomilla) grown in Belgium. Ind. Crop. Prod. 2010, 31, 145–152.
- Mertens D.; Boege K.; Kessler A.; Koricheva J.; Thaler J.S.; Whiteman N.K.; Poelman E.H. Predictability of biotic stress structures plant defence evolution. Ecol. Evol. 2021, 36, 444-456.
- Sarrou E.; Martens S.; Chatzopoulou P. Metabolite profiling and antioxidative activity of Sage (Salvia fruticosa) under the influence of genotype and harvesting period. Ind. Crop. Prod. 2016, 94, 240–250.
Reviewer 3 Report
The topic of the article is new and interesting, it meets the standards of the journal, and therefore it is suitable for publication in the journal. In the article, information on the Classification, distribution, biosynthesis, and regulation of secondary metabolites in Matricaria chamomilla is collected in one place. However, there are some things that would do well to correct. These are marked in the text.

Author Response
Dear Reviewer 3,
Thank you for your comments concerning our manuscript entitled “Classification, distribution, biosynthesis, and regulation of secondary metabolites in Matricaria chamomilla” (Manuscript ID: horticulturae-2008817).
Those comments are all valuable and very helpful for revising and improving our paper, as well as the important guiding significance to our work. We have studied comments carefully and have made correction which we hope meet with approval. Revised portion are marked in red in the paper. The main corrections in the paper and the responds to the reviewer’s comments are as following:
The topic of the article is new and interesting, it meets the standards of the journal, and therefore it is suitable for publication in the journal. In the article, information on the Classification, distribution, biosynthesis, and regulation of secondary metabolites in Matricaria chamomilla is collected in one place. However, there are some things that would do well to correct. These are marked in the text.
Response: I'm sorry that we made such a mistake. According to the reviewer's suggestion, we have changed to “Matricaria chamomilla is an ornamental medicinal aromatic plant of the composite family Asteraceae, widely distributed in Europe, Asia, the Mediterranean, southern Africa, and northwestern America. As a result of its unique fragrance and pharmacological effects, it is widely used in food, beverages, tobacco, daily chemicals, medicine, and other fields, with good market development potential for new agricultural plants[1]. There are two species of chamomile commonly used: Chamaemelum nobile and M. chamomilla, derived from the family Asteraceae[1-2]. Because the appearance of the two is very similar and the standard of medicinal materials is low, the C. nobile is often mistaken for the M. chamomilla in the process of purchasing medicinal materials. According to a large number of pharmacological studies on the volatile oil of M. chamomilla, the plant is a medicinal herb with antiseptic, anti-inflammatory, antispasmodic, and soothing effects [3].”
Response: I'm sorry that we made such a mistake. According to the reviewer's suggestion, we have made the following changes:
- Active constituents of M. chamomilla
The previous studies have found that M. chamomilla contains more than 120 types of medicinal active ingredients, including 56 types of organic acids 36 types of flavonoids, 28 types of terpenoidsand other compounds such as coumarins [5]. Yang and Pan [9] identified two flavonoids, lutein-7-O-β-D-glucoside (II) and apigenin-7-O-β-D-glucoside (I), in the inflorescence of M. chamomilla from Xinjiang. Zhou and Li [10] found that M. chamomilla contains quercetin, apigenin, luteolin, umbelliferone, and luteolin 7-O-β-d-glucoside (V). In addition, galactose, galacturonic acid, xylose, and choline were detected in M. chamomilla. Apigenin is an abundant water-soluble flavonoid in M. chamomilla. It mainly exists in the form of apigenin-7-O glucoside and other acylated derivatives [11].
As the main medicinal active ingredient source of volatile oil, the important substances are terpenoids. Monoterpenes and sesquiterpenes are the most abundant, of which oxygen-containing derivatives are less abundant but mostly have aroma. In recent years, researchers used GC–MS to analyze the volatile oil components of M. chamomilla. They found that the main components in the volatile oil of M. chamomilla are sesquiterpenoids, such as α-bisabolol, chamazulene, α-bisabolol oxide, farnesene, β-ocimene, α-elemene, α-pinene, and absinthol [12]. Sesquiterpene, flavonoids, coumarins, and polyacetylene are considered the most important substances in the medicinal active ingredients of M. chamomilla [13].
Many sentences are illegible, rewrite it.
Response: I'm sorry that due to our negligence, the meaning of this sentence is not accurate. It should be changed to “α-Bisabolol and its oxides only existed in buds and flowers, but their concentrations were negatively correlated.”
Saline irrigation also affects and changes the morphology, agronomic, phytochemical traits of M. chamomilla. Nitrogen is an essential substance for plant growth and metabolism [77]. Plants must balance the metabolism of carbon-rich metabolites under nitrogen limitation [78]. Phenylalanine ammonia-lyase (PAL) is a key enzyme in the biosynthesis of flavonoid, benzoic acid and coumarin, and its activity can limit the accumulation of phenolic secondary metabolites. Higher nitrogen concentration in the field may promote plant growth, increase chlorophyll content, and finally increase the content of phenolic metabolites with antioxidant activity such as umbelliferone in M. chamomilla [79].”
“which causes damage to roots, stems, flowers, and leaves, thus affecting the quality of essential oil [55]. “
“Raal et al. [84] found that the content of α-bisabolol oxide B was higher than that of α-bisabolol oxide A in M. chamomilla from Lativa and Poland, while the situation was opposite in M. chamomilla from Lithuania and Netherlands.”
“There were significant differences in volatile oil components of M. chamomilla from different seedling ages and different regions (Bodegold, Bona, Lutea, Germania, Manzana, Pnos, and Zloty LAN). The contents of sesquiterpene, β-caryophyllene, and β-farnesene were generally dominant in the first 30 days of seedling development of M. chamomilla, whereas other terpenes such as β-ocimene, citronella, epoxybutane, and germacrene D varied significantly during seedling growth [55]. No α-bisabolol and its oxides were found during this growth process, which contradicts Mohammad et al[86]. A large amount of sesquiterpenes that form at the beginning of the development of chamomile seedlings may minimize the production of reactive oxygen species under physiological conditions during seedling growth. M. chamomilla of Manzana and Pnos began to produce spiroethers after 30 days, while the other five locations continued to accumulate high levels in the first 40 days. In general, the young plant will show stronger defense ability than the mature plant, allowing the plant to adapt to different kinds of stress [87]. Total metabolite concentrations produced by chamomile seedlings in stems, leaves, buds or flowers were significantly lower than those of mature individuals. As the seedlings developed, the production of primary metabolite may be more concentrated upon growth and development, and the production of secondary metabolites is inhibited in the larval stage. In the study of Mondal [55], the complex terpene mixtures produced from the vegetative parts of M. chamomilla did not contain α-bisabolol, and there were significant differences in the synthesis of terpene metabolites. The variability of their metabolite production affected all ecological interactions. No matter which variety of M. chamomilla, α-bisabolol and its oxides are not present in the stems and leaves, and the content of secondary metabolites in young organs are greater than mature leaves or stems. This is consistent with the results of Sarrou et al. [88] investigating qualitative and quantitative differences in secondary metabolites of Salvia plants among populations of same species [55].
6.4 Metabolic differences of mature varieties from different regions
In general, the number of metabolites varies significantly between germplasms.”
References in the name of the magazine need full name. The format of the reference is incorrect
Response: Corrected accordingly.
The plant name should be in italic.
Response: I'm sorry that we made such a mistake. According to the reviewer's suggestion, we changed all the plant names to italics.
English language and style are fine/minor spell check required.
Response: I'm sorry that we made such a mistake. According to the reviewer's suggestion, we carefully examined the article and revised its grammar and spelling mistakes.

Reviewer 4 Report
The manusucript is well described, but in the references section should update some of the references that are so old.
Author Response
Dear Reviewers 4,
Thank you for your comments concerning our manuscript entitled “Classification, distribution, biosynthesis, and regulation of secondary metabolites in Matricaria chamomilla” (Manuscript ID: horticulturae-2008817).
Those comments are all valuable and very helpful for revising and improving our paper, as well as the important guiding significance to our work. We have studied comments carefully and have made correction which we hope meet with approval. Revised portion are marked in red in the paper. The main corrections in the paper and the responds to the reviewer’s comments are as following:
The manuscript is well described, but in the references, section should update some of the references that are so old.
Response: We are very grateful for your providing of the suggestions above. According to the reviewer's good instruction, we have updated the references in this paper. The modifications are as follows:
References
- Liu X.M.; Meng X.X.; Zhang W.W.; Liao Y.L.; Chang J.; Xu F. Tissue Culture Technology of Matricaria chamomilla North. Hortic. 2018, 2, 72-76.
- Han S.L.; Li X.X.; Mian Q.H.; Lan W.; Liu Y. Comparison of antioxidant activity between two species of chamomiles produced in Xinjiang by TLC-bioautography. China J. Chin. Mater. Med. 2013, 38, 193-198.
- Wan W.T.; Song Y.J.; Xu L.J.; Xiao P.G.; Miao J.H. An overview on modern research and application potency of Chamomile. Chin. Med. 2019, 21, 260-265.
- Bhattacharjee S.K. Handbook of aromatic plants. India: Pointer Publishers, 2005; pp. 277–279.
- Xia Q.X.; Bai H.T.; Sun L.C.; Gao T.G.; Jiang C.D.; Shi L. Research progress on active composition and practical application of medicinal plants of Matricaria recutita. Acta Hortic. Sinica 2012, 39, 1859-1864.
- Shiva M.P.; Lehri A.; Shiva A. Matricaria-Chamomilla/IAromatic and medicinal plants. International Book Distributors, India, 2002; pp. 223–228.
- Xu L. United States Pharmacopeia (24th Edition): Chamomile. World Notes Plant Med. 2002, 2, 82-83.
- Wu Z.Y. Flora of China. Beijing:Science Press 2010; pp. 49–50.
- Yang Y.S.; Pan L.S. Isolation and structural determination of flavones from Matricaria chamomilla Appl. Chem. Ind. 2008, 37, 697–698.
- Zhou B.T.; Li X.Z. Study of the chemical composition of chamomile. Hunan Coll. Tradit. Chin. Med. 2001, 21, 27–28.
- Zhao Y.F. Chemical constituents and quality standard of Matricaria chamomilla as uygur medicine. MA thesis, China Academy of Chinese Medical Sciences, Beijing, China, 2018.
- Krüger H. Characterisation of Chamomile Volatiles by Simultaneous Distillation Solid-Phase Extraction in Comparison to Hydrodistillation and Simultaneous Distillation Extraction. Planta Med. 2010, 76, 843–846.
- Schilcher H. Die Kamille: Handbuch für Ärzte,Apotheker und andere Naturwissenschaftler. Wissenschaftliche Verlagsgesellschaft, Stuttgart: Wiss. Verl, 1987; pp. 59–61.
- Kazemi M. Chemical Composition and Antimicrobial Activity of Essential Oil of Matricaria recutita. J. Food Prop. 2015, 18, 1784-1792.
- Wesolowska A.; Grzeszczuk M.; Kulpa D. Propagation Method and Distillation Apparatus Type Affect Essential Oil from Different Parts of Matricaria recutita Plants. J. Essent. Oil Bear. Pl. 2015, 18:179-194.
- Zhao Y.F.; Zhang D.; Liang X.X.; Yang L.X.; Sun P.; Ma Y.; Wang K.; Chang X.Q.; Yang L. Chemical constituents from Matricaria chamomilla(Ⅰ). J. Chin. Pharm. Sci. 2018, 27, 324-331.
- Cordero C.; Sgorbini B.; Rubiolo P.; Belliardo F.; Liberto E.; Bicchi C. Headspace–solid‐phase microextraction fast GC in combination with principal component analysis as a tool to classify different chemotypes of chamomile flower‐heads (Matricaria Recutita). Phytochem. Analysis 2006, 17, 217-225.
- Xu Y.B.; Tang H.; Zhu S.Y.; Zhe W.; Wang K.; Mao D.S.; Fu L.; Chen R.R. Analysis of Volatile Components of Chamomile Oil of Different Origins by Gas Chromatography Time-of-Flight Mass Spectrometry. Technol. Food Ind. 2015, 36, 6.
- Das M.; Ram G.; Singh A.; Mallavarapu G.R.; Ramesh S.; Ram M.; Kumar S. Volatile constituents of different plant parts of Chamomilla recutita Rausch grown in the Indo-Gangetic plains. Flavour. Fragr. J. 2002, 17, 9–12.
- Povh N.P.; Garcia C.A.; Marques M.O.M.; Meireles M.A.A. Extraction of essential oil and oleoresin from chamomile (Chamomila recutita [L.] Rauschert) by steam distillation and extraction with organic solvents: a process design approach. Bras. Pl. Med. 2001, 4, 1–8.
- McKay D.L.; Blimiberg J.B. A review of the bioactivity and potential health benefits of chamomile tea (Matricaria recutita). Phytother. Res. 2006, 20, 519–530.
- Russell K.; Jacob S.E. Bisabolol. Dermatitis 2010, 21, 57–58.
- Uno M.; Kokuryo T.; Yokoyama Y.; Senga T.; Nagino M. α-Bisabolol Inhibits Invasiveness and Motility in Pancreatic Cancer Through KISS1R Activation. Anticancer 2016, 36, 583–589.
- Yan H.B.; Xu R.X. Effect of α-bisabolol on migration and invasion of glioblastoma cells. J. Chin. Pla Med. Sch. 2018, 39, 699–706.
- Sharkey T.D.; Gray D.W.; Pell H.K.; Breneman S.R.; Topper L. Isoprene synthase genes form a monophyletic clade of acyclic terpene synthases in the TPS-B terpene synthase family. Int. J. Org. Evol. 2013, 67, 1026–1040.
- Schilcher H.; Imming P.; Goeters S. Active Chemical Constituents of Matricaria chamomilla syn. Chamomilla recutita (L.) Rauschert. Chamomile ind. profile. 2005, 1, 56–76.
- Farhoudi R. Chemical constituents and antioxidant properties of Matricaria Recutita and Chamaemelum nobile essential oil growing wild in the south west of Iran. Essent. Oil Bear. Pl. 2013, 16, 531–537.
- Schnee C.; Köllner T.G.; Held M.; Turlings T.C.J.; Gershenzon J.; Degenhardt J. The products of a single maize sesquiterpene synthase form a volatile defense signal that attracts natural enemies of maize herbivores. Natl. A. Sci. 2006, 103, 1129–1134.
- Köllner T.G.; Held M.; Lenk C.; Hiltpold I.; Turlings T.C.; Gershenzon J.; Degenhardt J. A Maize (E)-β-Caryophyllene Synthase Implicated in Indirect Defense Responses against Herbivores Is Not Expressed in Most American Maize Plant Cell 2008, 20, 482–494.
- Yu L.T.; Cheng C.L.; Cheng X.W.; Huan H.W.; Ling S.; Lin Y.; Wei J.; Xiao R.Y.; Lu J.Z.; Zhan F.; Chun L.; Yi Y. Analysis of terpenoid biosynthesis pathways in german chamomile (Matricaria recutita) and roman chamomile (Chamaemelum nobile) based on co-expression networks. Genomics 2020, 112, 1055-1064.
- Wright G.A.; Schiestl F.P. The evolution of floral scent: the influence of olfactory learning by insect pollinators on the honest signalling of floral rewards. Ecol. 2009, 23, 841–851.
- Zhang W.W.; Tao T.T.; Liu X.M.; Xu F.; Chang J.; Liao Y.L. De novo assembly and comparative transcriptome analysis: novel insights into sesquiterpenoid biosynthesis in Matricaria chamomilla Acta Physiol. Plant. 2018, 40, 1–14.
- Han X.; Wang Y.D.; Chen Y.C.; Lin L. Y.; Wu Q.K.; Christian S. Transcriptome Sequencing and Expression Analysis of Terpenoid Biosynthesis Genes in Litsea cubeba. PloS One 2013, 8: e76890.
- Zhao Y.J.; Chen X.; Zhang M.; Ping S.; Liu Y.J.; Tong Y.R.; Wang X.J.; Huang L.Q.; Gao W. Molecular cloning and characterisation of farnesyl pyrophosphate synthase from Tripterygium wilfordii. PloS One 2015, 10, e0125415.
- Ortiz de Montellano P.R. Acetylenes: cytochrome P450 oxidation and mechanism-based enzyme inactivation. Drug Metab. Rev. 2019, 51, 162–177.
- Ling S.P.; Zhang H.M.; Su S.S.; Zhang X.S.; Liu X.Y.; Pan G.F.; Yuan Y. Molecular Cloning and characterization of a Cytochrome P450 reductase gene (CPR) full-length in Matricaria recutita. Agr. Biotechnol. 2014, 22, 580–589.
- Fu M.; Liu X.; Meng X.; Wang L.; Tan J.; Zhou X.; Xu F. Cloning and Sequence Analysis of an Acetyl-CoA C-Acetyltransferase Gene (AACT) from Chamaemelum nobile. J. Curr. Res. Biosci. Plant Biology 2017, 4, 31–37.
- Yan J.; Meng; Xu F.; Chang J. Molecular cloning and sequence analysis of a phosphomevalonate kinase gene (CnPMK) from Chamaemelum nobile. Int. J. Curr. Res. Biosci. Plant Biology 2016, 3,157–162.
- Meng X.X.; Zhang W.; Xu F.; Yan J.; Liu X.; Liao Y.L.; Chang J. Cloning and sequence analysis of mevalonate kinase gene (CnMVK) from Chamaemelum nobile. J. Curr. Res. Biosci. Plant Biology 2016, 3, 23–28.
- Cheng S.Y.; Wang X.H.; Xu F.; Chen Q.W.; Tao T.T.; Lei J.; Zhang W.W.; Liao Y.L.; Chang J.; Li X.X.; Gulder T.A.M. Cloning, Expression Profiling and Functional Analysis of CnHMGS, a Gene Encoding 3-hydroxy-3-Methylglutaryl Coenzyme A Synthase from Chamaemelum nobile. Molecules 2016, 21, 316.
- SuS.; Liu X.Y.; Pan G.F.; Hou X.J.; Zhang H.M.; Yuan Y. In vitro characterization of a (E)-β-farnesene synthase from Matricaria recutita L. and its up-regulation by methyl jasmonate. Gene 2015, 571, 58–64.
- Tao T.T.; Chen Q.W.; Meng X.X.; Yan J.P.; Xu F.; Chang J. Molecular cloning, characterization, and functional analysis of a gene encoding 3-hydroxy-3-methylglutaryl-coenzyme A synthase from Matricaria chamomilla. Genes Genom. 2016, 38, 1179–1187.
- Su S.S.; Zhang H.M.; Liu X.Y.; Pan G.F.; Ling S.P.; Zhang X.S.; Yang X.M.; Tai Y.L.; Yuan Y. Cloning and characterization of a farnesyl pyrophosphate synthase from Matricaria recutita and its upregulation by methyl jasmonate. Genet. Mol. Res. 2015, 14, 349–361.
- Meng X.X.; Yan J.P.; Liao Y.L.; Chang J.; Xu F. Cloning and expression analysis of HMGR gene from Chamaemelum nobile. Acta Agr. Boreal. Sinica 2016, 31, 68–75.
- Sun J.M. Omics-based study on genes related to volatile terpenoids from Matricaria chamomilla MA thesis. Anhui Agricultural University, Anhui, China, 2018.
- Tai Y.L. Molecular Cloning, Expression of FPS and analysis of salt stress in Matrucarua chamomilla MA thesis, Anhui Agricultural University, Anhui, China, 2012.
- Yang X.M. Study on Rapid Propagation of Matricaria Chamomile L and genetic transformation of FPS gene in Tobacco. MA thesis, Anhui Agricultural University, Anhui, China, 2013.
- Gupta P.; Akhtar N.; Tewari S.K.; Sangwan R.S.; Trivedi P.K. Differential expression of farnesyl diphosphate synthase gene from Withania somnifera in different chemotypes and in response to elicitors. Plant Growth Regul. 2011, 65, 93–100.
- Lan J.B.; Yu R.C.; Yu Y.Y.; Fan Y.P. Molecular cloning and expression of Hedychium coronarium farnesyl pyrophosphate synthase gene and its possible involvement in the biosynthesis of floral and wounding/herbivory induced leaf volatile sesquiterpenoids. Gene 2013, 518, 360–367.
- Xiang L.; Zhao K.; Chen L.Q. Molecular cloning and expression of Chimonanthus praecox farnesyl pyrophosphate synthase gene and its possible involvement in the biosynthesis of floral volatile sesquiterpenoids. Plant Physiol. Bioch. 2010, 48, 845–850.
- Su S.S. Cloning and characterization of (E)-β-farnesene synthase from Matricaria recutita MA thesis, Anhui Agricultural University, Anhui, China, 2015.
- Son Y.J.; Kwon M.; Ro D.K.; Kim S.U. Enantioselective microbial synthesis of the indigenous natural product (−)-α-bisabolol by a sesquiterpene synthase from chamomile (Matricaria recutita). J. 2014, 463, 239–248.
- Irmisch S.; Krause S.T.; Kunert G.; Gershenzon J.; Degenhardt J.; Köllner T.G. The organ-specific expression of terpene synthase genes contributes to the terpene hydrocarbon composition of chamomile essential oils. BMC Plant Biol. 2012, 12, 1–13.
- Guo C.X. Cloning and characterization of α-bisabolol synthase gene (MrBBS) from Matricaria chamomilla MA thesis, Anhui Agricultural University, Anhui, China, 2018.
- Mondal P. Biosynthesis and regulation of terpene production in accessions of chamomile (Matricaria recutita). Ph D thesis, Martin-Luther-Universität Halle-Wittenberg, Halle, Germany, 2020.
- Ling C.C.; Zheng L.J.; Yu X.R.; Wang H.H.; Wang C.X.; Wu H.Y.; Zhang J.; Yao P.; Tai Y.L.; Yuan Y. Cloning and functional analysis of three aphid alarm pheromone genes from German chamomile (Matricaria chamomilla). Plant Sci. 2020, 294, 110463.
- Yan J.P.; Meng X.X.; Zhu L.; Zhang W.W.; Chang J.; Xu F. Molecular cloning and expression analysis of germacrene A synthase gene in Chamaemelum nobile. Tradit. Herb. Drugs 2017, 48, 1851–1859.
- Ling S.P. Molecular Cloning and characterization of Squalene synthase gene and analysis of essential oil from Different organs of Matricaria recutita. MA thesis, Anhui Agricultural University, Anhui, China, 2014.
- Liu X.M.; Tao T.T.; Meng X.X.; Zhang W.W.; Chang J.; Xu F. Cloning and expression analysis of a farnesyl diphosphate synthase (FPPS) gene from Chamaemelum nobile. Bot. Horti Agrobo. 2017, 45, 358–364.
- Zhang T.; Li Z.Q.; Wu G.Q. Role of WRKY Transcription Factor in Plant Response to Stresses. Bull. 2021, 37, 203-215.
- Xiang L.; Zhang W.W.; Qu J.W.; Han H.; Xu F.; Liao Y.L. Identification and Expression Analysis of the WRKY Transcription Factor in Matricaria recutita Mol. Plant Breeding 2020, 18, 2127–2137.
- Izumi S.; Takashima O.; Hirata T. Geraniol Is a Potent Inducer of Apoptosis-like Cell Death in the Cultured Shoot Primordia of Matricaria chamomilla. Bioph. Res. Co. 1999, 259, 519–522.
- Ashida Y.; Nishimoto M.; Matsushima A.; Watanabe J.; Hirata T. Molecular Cloning and mRNA Expression of Geraniol-inducible Genes in Cultured Shoot Primordia of Matricaria chamomilla. Biotech. Bioch. 2002, 66, 2511–2514.
- Li Y.Y. The main composition and affecting factors of aroma volatiles in flowers. Northern Horticult. 2012, 6, 184-189.
- Sharifi-Rad M; Nazaruk J; Polito L; Morais-Braga M.F.B.; Rocha J.E.; Coutinho H.D.M.; Salehi B.; Tabanelli G.; Montanari C.; Del Mar Contreras M.; Yousaf Z.; Setzer W.N.; Verma D.R.; Martorell M.; Sureda A.; Sharifi-Rad J. Matricaria genus as a source of antimicrobial agents: From farm to pharmacy and food applications. Res. 2018, 215, 76-88.
- Sagi S.; Avula B.; Wang Y.H.; Zhao J.; Khan I.A. Quantitative determination of seven chemical constituents and chemo-type differentiation of chamomiles using high-performance thin-layer chromatography. J. Sep. Sci. 2014. 37, 2797-804.
- Mobasheri L.; Khorashadizadeh M.; Safarpour H.; Mohammadi M.; Anani Sarab G.; Askari V.R. Anti-inflammatory activity of ferula assafoetida Oleo-Gum-Resin (Asafoetida) against TNF-α-stimulated Human Umbilical Vein Endothelial Cells (HUVECs). Inflamm. 2022. 31, 5171525.
- Avallone R.; Zanoli P.; Puia G.; Kleinschnitz M.; Schreier P.; Baraldi M. Pharmacological profile of apigenin, a flavonoid isolated from Matricaria chamomilla. Pharmacol. 2000, 59, 1387–1394.
- Švehlíková V.; Bennett R.N.; Mellon F.A.; Needs P.W.; Piacente S.; Kroon P.A.; Bao Y. Isolation, identification and stability of acylated derivatives of apigenin 7-O-glucoside from chamomile (Chamomilla recutita [L.] Rauschert). Phytochemistry 2004, 65, 2323–2332.
- Babenko N.A, Shakhova E.G. Effects of Chamomilla recutita flavonoids on age-related liver sphingolipid turnover in rats. Gerontol. 2006, 41, 32–39.
- Weber B.; Herrmann M.; Hartmann B.; Joppe H.; Schmidt C.O.; Bertram H.J. HPLC/MS and HPLC/NMR as hyphenated techniques for accelerated characterization of the main constituents in Chamomile (Chamomilla recutita [L.] Rauschert). Food Res. Technol. 2008, 226, 755–760.
- Cheng S.Y.; Song Q.; Tian Y.; Liu X.; Wang L.; Mao D.; Zhang W.; Xu F. Characterization and expression analysis of four members genes of flavanone 3-hydroxylase families from chamaemelum nobile. Bot. Horti Agrobo. 2020, 48, 102–115.
- Badi H.N.; Yazdani D.; Ali S.M.; Nazari F. Effects of spacing and harvesting time on herbage yield and quality/quantity of oil in thyme, Thymus vulgaris L. Crop. Prod. 2004, 19, 231–236.
- Banchio E.; Zygadlo J.; Valladares G. R. Quantitative variations in the essential oil of Minthostachys mollis (kunth.) griseb. in response to insects with different feeding habits. Agr. Food Chem. 2005, 53, 6903–6906.
- Ghasemi M.; Babaeian J. N.; Modarresi M.; Bagheri N.; Jamali A. Increase of Chamazulene and α-Bisabolol Contents of the Essential Oil of German Chamomile (Matricaria chamomilla) Using Salicylic Acid Treatments under Normal and Heat Stress Conditions. Foods 2016, 5, 56.
- Liu X.; Zhu L.; Song Q.; Chang J.; Ye J.; Zhang W.; Liao Y.; Xu F. Effects of 5-aminolevulinic Acid on the Photosynthesis, Antioxidant System, and α-Bisabolol Content of Matricaria recutita. Bot. Horti Agrobo. 2018, 46, 418–425.
- Kusano M.; Fukushima A.; Redestig H.; Saito K. Metabolomic approaches toward understanding nitrogen metabolism in plants. Exp. Bot. 2011, 62, 1439–1453.
- Matt P.; Krapp A.; Haake V.; Mock H.P.; Stitt M. Decreased Rubisco activity leads to dramatic changes of nitrate metabolism, amino acid metabolism and the levels of phenylpropanoids and nicotine in tobacco antisense RBCS transformants. Plant J. 2002, 30, 663–677.
- Repčák M.; Imrich J.; Franeková M. Umbelliferone, a stress metabolite of Chamomilla recutita (L.) Rauschert. Plant Physiol. 2001, 158, 1085–1087.
- Maksymiec W. Signaling responses in plants to heavy metal stress. Acta Physiol. Plant. 2007, 29, 177–187.
- Kováčik J.; Tomko J.; Bačkor M.; Repčák M. Matricaria chamomilla is not a hyperaccumulator, but tolerant to cadmium stress. Plant Growth Regul. 2006, 50, 239–247.
- Prokop’ev I.A.; Filippova G.V.; Shein A.A.; Gabyshev D.V. Impact of urban anthropogenic pollution on seed production, morphological and biochemical characteristics of chamomile, Matricaria chamomila Russ. J. Ecol. 2014, 45, 18–23.
- Kessler A.; Baldwin I.T. Plant responses to insect herbivory: the emerging molecular analysis. Rev. Plant Biol. 2002, 53, 299–328.
- Raal A.; Kaur H.; Orav A.; Arak E.; Kailas T.; Muurisepp M. Content and composition of essential oils in some Asteraceae species. Natl A. Sci. 2011, 60, 55–63.
- Orav A.; Kailas T.; Ivask K. Volatile constituents of Matricaria recutita from Estonia. Est. A. Sci. Chem. 2001, 50, 39–45.
- Mohammad R.; Hamid S.; An A.; Norbert D.K.; Patrick V.D. Effects of planting date and seedling age on agro-morphological characteristics, essential oil content and composition of German chamomile (Matricaria chamomilla) grown in Belgium. Ind. Crop. Prod. 2010, 31, 145–152.
- Mertens D.; Boege K.; Kessler A.; Koricheva J.; Thaler J.S.; Whiteman N.K.; Poelman E.H. Predictability of biotic stress structures plant defence evolution. Ecol. Evol. 2021, 36, 444-456.
- Sarrou E.; Martens S.; Chatzopoulou P. Metabolite profiling and antioxidative activity of Sage (Salvia fruticosa) under the influence of genotype and harvesting period. Ind. Crop. Prod. 2016, 94, 240–250.

Round 2
Reviewer 1 Report
The authors have corrected and improved the manuscript thoroughly. Therefore, I believe the manuscript in its current form is suitable for the publication in Horticulturae journal.
Author Response
We thank the reviewer for the encouraging comments.
Reviewer 3 Report
The manuscript, after the revision is accepted for publication, however, there are some things that would do well to correct. These are marked in the text.

Author Response
Dear Reviewer 3,
Thank you for your comments concerning our manuscript entitled “Classification, distribution, biosynthesis, and regulation of secondary metabolites in Matricaria chamomilla” (Manuscript ID: horticulturae-2008817).
Those comments are all valuable and very helpful for revising and improving our paper, as well as the important guiding significance to our work. We have studied comments carefully and have made correction which we hope meet with approval. Revised portion are marked in blue in the paper. The main corrections in the paper and the responds to the reviewer’s comments are as following.